# Random Matrix Theory Proves that Deep Learning Representations of GAN-data Behave as Gaussian Mixtures

## Abstract

This paper shows that deep learning (DL) representations of data produced by generative adversarial nets (GANs) are random vectors which fall within the class of so-called *concentrated* random vectors. Further exploiting the fact that Gram matrices, of the type $\boldsymbol{G} = \boldsymbol{X}^\intercal \boldsymbol{X}$ with $\boldsymbol{X} = [\boldsymbol{x}_1, \ldots, \boldsymbol{x}_n] \in \mathbb{R}^{p \times n}$ and $\boldsymbol{x}_i$ independent concentrated random vectors from a mixture model, behave asymptotically (as $n, p \to \infty$) as if the $\boldsymbol{x}_i$ were drawn from a Gaussian mixture, suggests that DL representations of GAN-data can be fully described by their first two statistical moments for a wide range of standard classifiers. Our theoretical findings are validated by generating images with the BigGAN model and across different popular deep representation networks.

## 1 Introduction

The performance of machine learning methods depends strongly on the choice of the data representation (or features) on which they are applied. This data representation should ideally contain *relevant information* about the learning task in order to achieve learning with *simple* models and *small* amount of samples. Deep neural networks (Rumelhart et al., 1988) have particularly shown impressive results by automatically learning representations from raw data (*e.g.*, images). However, due to the complex structure of deep learning models, the characterization of their hidden representations is still an open problem (Bengio et al., 2009).

Specifically, quantifying what makes a given deep learning representation better than another is a fundamental question in the field of *Representation Learning* (Bengio et al., 2013). Relying on (Montavon et al., 2011) a data representation is said to be *good* when it is possible to build *simple* models on top of it that are *accurate* for the given learning problem. Montavon et al. (2011) have notably quantified the layer-wise evolution of the representation in deep networks by computing the principal components of the Gram matrix $\boldsymbol{G}_\ell = \{\phi_\ell(\boldsymbol{x}_i)^\intercal \phi_\ell(\boldsymbol{x}_j)\}_{i,j=1}^n$ at each layer for $n$ input data $\boldsymbol{x}_1, \ldots, \boldsymbol{x}_n$, where $\phi_\ell(\boldsymbol{x})$ is the representation of $\boldsymbol{x}$ at layer $\ell$ of the given DL model, and the number of components controls the model simplicity. In their study, the impact of the representation at each layer is quantified through the prediction error of a linear predictor trained on the principal subspace of $\boldsymbol{G}_\ell$.

Pursuing on this idea, given a certain representation model $\boldsymbol{x} \mapsto \phi(\boldsymbol{x})$, we aim in this article at theoretically studying the large dimensional behavior, and in particular the spectral information (*i.e.*, eigenvalues and dominant eigenvectors), of the corresponding Gram matrix $\boldsymbol{G} = \{\phi(\boldsymbol{x}_i)^\intercal \phi(\boldsymbol{x}_j)\}_{i,j=1}^n$ in order to determine the information encoded (*i.e.*, the sufficient statistics) by the representation model on a set of real data $\boldsymbol{x}_1, \ldots, \boldsymbol{x}_n$. Indeed, standard classification and regression algorithms –along with the last layer of a neural network (Yeh et al., 2018)– retrieve the data information directly from functionals or the eigenspectrum of $\boldsymbol{G}$[1]. To this end, though, one needs a statistical model for the representations given the distribution of the raw data (*e.g.*, images) which is generally unknown. Yet, due to recent advances in generative models since the advent of Generative Adversarial Nets (Goodfellow et al., 2014), it is now possible to generate complex data

---

[1]For instance, spectral clustering uses the dominant eigenvectors of $\boldsymbol{G}$, while support vector machines use functionals (quadratic forms) involving $\boldsymbol{G}$.

structures by applying successive *Lipschitz* operations to Gaussian random vectors. In particular, GAN-data *are* used in practice as substitutes of real data for data augmentation (Antoniou et al., 2017). On the other hand, the fundamental concentration of measure phenomenon (Ledoux, 2005) tells us that Lipschitz-ally transformed Gaussian vectors satisfy a concentration property. Precisely, defining the class of *concentrated* vectors $x \in E$ through concentration inequalities of $f(x)$, for any real Lipschitz observation $f : E \to \mathbb{R}$, implies that deep learning representations of GAN-data fall within this class of random vectors, since the mapping $x \mapsto \phi(x)$ *is* Lipschitz. Thus, GAN-data are concentrated random vectors and thus an appropriate statistical model of realistic data.

Targeting classification applications by assuming a mixture of concentrated random vectors model, this article studies the spectral behavior of Gram matrices $G$ in the large $n, p$ regime. Precisely, we show that these matrices have asymptotically (as $n, p \to \infty$ with $p/n \to c < \infty$) the same first-order behavior as for a Gaussian Mixture Model (GMM). As a result, by generating images using the BigGAN model (Brock et al., 2018) and considering different commonly used deep representation models, we show that the spectral behavior of the Gram matrix computed on these representations is the same as on a GMM model with the same $p$-dimensional means and covariances. A surprising consequence is that, for GAN data, the aforementioned *sufficient statistics* to characterize the quality of a given representation network are only the *first* and *second* order statistics of the representations. This behavior is shown by simulations to extend beyond random GAN-data to real images from the Imagenet dataset (Deng et al., 2009).

The rest of the paper is organized as follows. In Section 2, we introduce the notion of concentrated vectors and their main properties. Our main theoretical results are then provided in Section 3. In Section 4 we present experimental results. Section 5 concludes the article.

*Notation:* In the following, we use the notation from (Goodfellow et al., 2016). $[n]$ denotes the set $\{1, \ldots, n\}$. Given a vector $x \in \mathbb{R}^n$, the $\ell_2$-norm of $x$ is given as $\|x\|^2 = \sum_{i=1}^{n} x_i^2$. Given a $p \times n$ matrix $M$, its Frobenius norm is defined as $\|M\|_F^2 = \sum_{i=1}^{p} \sum_{j=1}^{n} M_{ij}^2$ and its spectral norm as $\|M\| = \sup_{\|x\|=1} \|Mx\|$. $\odot$ for the Hadamard product. An application $\mathcal{F} : E \to F$ is said to be $\|\mathcal{F}\|_{lip}$-Lipschitz, if $\forall (x, y) \in E^2$, $\|\mathcal{F}(x) - \mathcal{F}(y)\|_F \leq \|\mathcal{F}\|_{lip} \cdot \|x - y\|_E$ and $\|\mathcal{F}\|_{lip}$ is finite.

## 2 BASIC NOTIONS OF CONCENTRATED VECTORS

Being the central tool of our study, we start by introducing the notion of concentrated vectors. While advanced concentration notions have been recently developed in (Louart & Couillet, 2019) in order to specifically analyze the behavior of large dimensional sample covariance matrices, for simplicity, we restrict ourselves here to the sufficient so-called $q$-exponentially concentrated random vectors.

**Definition 2.1** ($q$-exponential concentration). *Given a normed space $(E, \| \cdot \|_E)$ and a real $q$, a random vector $x \in E$ is said to be $q$-exponentially concentrated if for any 1-Lipschitz real function $f : E \to \mathbb{R}$, there exists $C \geq 0$ independent of $\dim(E)$ and $\sigma > 0$ such that for all $t \geq 0$*

$$\mathbb{P}\{|f(x) - \mathbb{E}f(x)| > t\} \leq C e^{-(t/\sigma)^q} \tag{1}$$

*which we denote $x \in \mathcal{E}_q(\sigma \,|\, E, \| \cdot \|_E)$. We simply write $x \in \mathcal{E}_q(1 \,|\, E, \| \cdot \|_E)$ if the tail parameter $\sigma$ does not depend on $\dim(E)$, and $x \in \mathcal{E}_q(1)$ for $x$ a scalar real random variable.*

Therefore, concentrated vectors are defined through the concentration of any 1-Lipschitz real scalar "observation". One of the most important examples of concentrated vectors are standard Gaussian vectors. Precisely, we have the following proposition. See (Ledoux, 2005)) for more examples such as uniform and Gamma distribution.

**Proposition 2.2** (Gaussian vectors (Ledoux, 2005)). *Let $d \in \mathbb{N}$ and $x \sim \mathcal{N}(0, I_d)$. Then $x$ is a 2-exponentially concentrated vector independently on the dimension $d$, i.e. $x \in \mathcal{E}_2(1 \,|\, \mathbb{R}^d, \| \cdot \|)$.*

Concentrated vectors have the interesting property of being stable by application of $\mathbb{R}^d \to \mathbb{R}^p$ vector-Lipschitz transformations. Indeed, Lipschitz-ally transformed concentrated vectors remain concentrated according to the following proposition.

**Proposition 2.3** (Lipschitz stability (Louart & Couillet, 2019)). *Let $x \in \mathcal{E}_q(1 \,|\, E, \| \cdot \|_E)$ and $\mathcal{G} : E \to F$ a Lipschitz application with Lipschitz constant $\|\mathcal{G}\|_{lip}$ which may depend on $\dim(F)$. Then the concentration property on $x$ is transferred to $\mathcal{G}(x)$, precisely*

$$x \in \mathcal{E}_q(1 \,|\, E, \| \cdot \|_E) \implies \mathcal{G}(x) \in \mathcal{E}_q(\|\mathcal{G}\|_{lip} \,|\, F, \| \cdot \|_F). \tag{2}$$

Note importantly for the following that the Lipschitz constant of the transformation $\mathcal{G}$ must be controlled, in order to constrain the tail parameter of the obtained concentration.

In particular, we have the coming corollary to Proposition 2.3 of central importance in the following.

**Corollary 2.4.** *Let $\mathcal{G}_1, \ldots, \mathcal{G}_n : \mathbb{R}^d \to \mathbb{R}^p$ a set of $n$ Lipschitz applications with Lipschitz constants $\|\mathcal{G}_i\|_{lip}$. Let $\mathcal{G} : \mathbb{R}^{d \times n} \to \mathbb{R}^{p \times n}$ be defined for each $\boldsymbol{X} \in \mathbb{R}^{d \times n}$ as $\mathcal{G}(\boldsymbol{X}) = [\mathcal{G}_1(\boldsymbol{X}_{:,1}), \ldots, \mathcal{G}_n(\boldsymbol{X}_{:,n})]$. Then,*

$$\boldsymbol{Z} \in \mathcal{E}_q(1 \,|\, \mathbb{R}^{d \times n}, \|\cdot\|_F) \;\Rightarrow\; \mathcal{G}(\boldsymbol{Z}) \in \mathcal{E}_q\left(\sup_i \|\mathcal{G}_i\|_{lip} \,|\, \mathbb{R}^{p \times n}, \|\cdot\|_F\right). \tag{3}$$

*Proof.* This is a consequence of Proposition 2.3 since the map $\mathcal{G}$ is $\sup_i \|\mathcal{G}_i\|_{lip}$-Lipschitz with respect to (w.r.t.) the Frobenius norm. Indeed, for $\boldsymbol{X}, \boldsymbol{H} \in \mathbb{R}^{d \times n}$ : $\|\mathcal{G}(\boldsymbol{X} + \boldsymbol{H}) - \mathcal{G}(\boldsymbol{X})\|_F^2 \leq \sum_{i=1}^n \|\mathcal{G}_i\|_{lip}^2 \cdot \|\boldsymbol{H}_{:,i}\|^2 \leq \sup_i \|\mathcal{G}_i\|_{lip}^2 \cdot \|\boldsymbol{H}\|_F^2.$ □

## 3 MAIN RESULTS

### 3.1 GAN DATA: AN EXAMPLE OF CONCENTRATED VECTORS

Concentrated random vectors are particularly interesting from a practical standpoint for real data modeling. In fact, unlike simple Gaussian vectors, the former do not suffer from the constraint of having independent entries which is quite a restrictive assumption when modeling real data such as images or their non-linear features (*e.g.*, DL representations). The other modeling interest of concentrated vectors lies in their being already present in practice as alternatives to real data. Indeed, adversarial neural networks (GANs) have the ability nowadays to generate random *realistic* data (for instance realistic images) by applying successive Lipschitz operations to standard Gaussian vectors (Goodfellow et al., 2014).

A GAN architecture involves two networks, a generator model which maps random Gaussian noise to new plausible synthetic data and a discriminator model which classifies real data as real (from the dataset) or fake (for the generated data). The discriminator is updated directly through a binary classification problem, whereas the generator is updated through the discriminator. As such, the two models are trained alternatively in an adversarial manner, where the generator seeks to better deceive the discriminator and the former seeks to better identify the fake data (Goodfellow et al., 2014).

In particular, once both models are trained (when they reach a Nash equilibrium), DL representations of GAN-data –and GAN-data themselves– are schematically constructed in practice as follows:

$$\text{Real Data} \approx \text{GAN Data} = \mathcal{F}_N \circ \cdots \circ \mathcal{F}_1(\boldsymbol{z}), \text{ where } \boldsymbol{z} \sim \mathcal{N}(0, \boldsymbol{I}_d), \tag{4}$$

where $d$ stands for the input dimension of the generator model, $N$ the number of layers, and the $\mathcal{F}_i$'s either Fully Connected Layers, Convolutional Layers, Pooling Layers, Up-sampling Layers and Activation Functions, Residual Layers or Batch Normalizations. All these operations happen to be *Lipschitz* applications. Precisely,

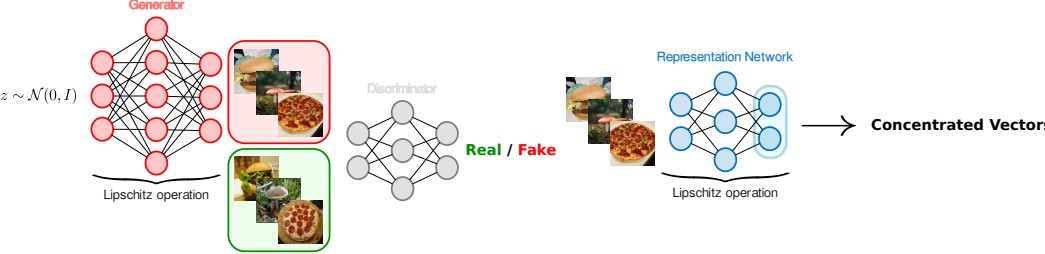

Figure 1: Deep learning representations of GAN-data are constructed by applying successive Lipschitz operations to Gaussian vectors, therefore they are *concentrated* vectors by design, since Gaussian vectors are concentrated and thanks to the Lipschitz stability in Proposition 2.3.

- **Fully Connected Layers and Convolutional Layers:** These are affine operations which can be expressed as

$$\mathcal{F}_i(\boldsymbol{x}) = \boldsymbol{W}_i\boldsymbol{x} + \boldsymbol{b}_i, \text{ for } \boldsymbol{W}_i \text{ the weight matrix and } \boldsymbol{b}_i \text{ the bias vector.}$$

Here the Lipschitz constant is the operator norm (the largest singular value) of the weight matrix $\boldsymbol{W}_i$, that is $\|\mathcal{F}_i\|_{lip} = \sup_{\boldsymbol{u}\neq 0} \frac{\|\boldsymbol{W}_i\boldsymbol{u}\|_2}{\|\boldsymbol{u}\|_2}$.

- **Pooling Layers and Activation Functions:** Most commonly used activation functions and pooling operations are

$$\text{ReLU}(\boldsymbol{x}) = \max(0, \boldsymbol{x}), \text{ MaxPooling}(\boldsymbol{x}) = [\max(\boldsymbol{x}_{\mathcal{S}_1}), \ldots, \max(\boldsymbol{x}_{\mathcal{S}_q})]^{\mathsf{T}},$$

where $\mathcal{S}_i$'s are patches (*i.e.*, subsets of $[\dim(\boldsymbol{x})]$). These are at most 1-Lipschitz operations with respect to the Frobenius norm. Specifically, the maximum absolute sub-gradient of the ReLU activation function is 1, thus the ReLU operation has a Lipschitz constant of 1. Similarly, we can show that the Lipschitz constant of MaxPooling layers is also 1.

- **Residual Connections:** Residual layers act the following way

$$\mathcal{F}_i(\boldsymbol{x}) = \boldsymbol{x} + \mathcal{F}_i^{(1)} \circ \cdots \circ \mathcal{F}_i^{(\ell)}(\boldsymbol{x}),$$

where the $\mathcal{F}_i^{(j)}$'s are Fully Connected Layers or Convolutional Layers with Activation Functions, and which are Lipschitz operations. Thus $\mathcal{F}_i$ is a Lipschitz operation with Lipschitz constant bounded by $1 + \prod_{j=1}^{\ell} \|\mathcal{F}_i^{(j)}\|_{lip}$.

- **Batch Normalization (BN) Layers:** They consist in statistically standardizing (Ioffe & Szegedy, 2015) the vectors of a small batch $\mathcal{B} = \{\boldsymbol{x}_i\}_{i=1}^b \subset \mathbb{R}^d$ as follows: for each $\boldsymbol{x}_k \in \mathcal{B}$

$$\mathcal{F}_i(\boldsymbol{x}_k) = \mathbf{diag}\left(\frac{\mathbf{a}}{\sqrt{\sigma_{\mathcal{B}}^2 + \epsilon}}\right)(\boldsymbol{x}_k - \mu_{\mathcal{B}}\mathbf{1}_d) + \mathbf{b}$$

where $\mu_{\mathcal{B}} = \frac{1}{db}\sum_{k=1}^b\sum_{i=1}^d[\boldsymbol{x}_k]_i$, $\sigma_{\mathcal{B}}^2 = \frac{1}{db}\sum_{k=1}^b\sum_{i=1}^d([\boldsymbol{x}_k]_i - \mu_{\mathcal{B}})^2$, $\boldsymbol{a}, \boldsymbol{b} \in \mathbb{R}^d$ are parameters to be learned and $\mathbf{diag}(\boldsymbol{v})$ transforms a vector $\boldsymbol{v}$ to a diagonal matrix with its diagonal entries being those of $\boldsymbol{v}$. Thus BN is a Lipschitz transformation with Lipschitz constant $\|\mathcal{F}_i\|_{lip} = \sup_i |\frac{\mathbf{a}_i}{\sqrt{\sigma_{\mathcal{B}}^2 + \epsilon}}|$.

Therefore, as illustrated in Figure 1, since standard Gaussian vectors are concentrated vectors as mentioned in Proposition 2.2 and since the notion of concentrated vectors is stable by Lipschitz transformations thanks to Proposition 2.3, GAN-data (and their DL representations) are concentrated vectors by design given the construction in Equation (4). Moreover, in order to generate data belonging to a specific class, Conditional GANs have been introduced (Mirza & Osindero, 2014); once again data generated by these models are concentrated vectors as a consequence of Corollary 2.4. Indeed, a generator of a Conditional GAN model can be seen as a set of multiple generators where each generates data of a specific class conditionally on the class label (*e.g.*, BigGAN model (Brock et al., 2018)).

Yet, in order to ensure that the resulting Lipschitz constant of the combination of the above operations does not scale with the network or data size, so to maintain good concentration behaviors, a careful control of the learned network parameters is needed. This control happens to be already considered in practice in order to ensure the stability of GANs during the learning phase, notably to generate realistic and high-resolution images (Roth et al., 2017; Brock et al., 2018). The control of the Lipschitz constant of representation networks is also needed in practice in order to make them robust against adversarial examples (Szegedy et al., 2013; Gulrajani et al., 2017). This control is particularly ensured through spectral normalization of the affine layers (Brock et al., 2018), such as Fully Connected Layers, Convolutional Layers and Batch Normalization. Indeed, spectral normalization (Miyato et al., 2018) consists in applying the operation $\boldsymbol{W} \leftarrow \boldsymbol{W}/\sigma_1(\boldsymbol{W})$ to the affine layers at each backward iteration of the back-propagation algorithm, where $\sigma_1(\boldsymbol{W})$ stands for the largest singular value of the weight matrix $\boldsymbol{W}$. Brock et al. (2018), have notably observed that, without spectral constraints, a subset of the generator layers grow throughout their GAN training and explode at collapse. They thus suggested the following spectral normalization –which happens to be less restrictive than the standard spectral normalization $\boldsymbol{W} \leftarrow \boldsymbol{W}/\sigma_1(\boldsymbol{W})$ (Miyato et al., 2018)– to the affine layers:

$$\boldsymbol{W} \leftarrow \boldsymbol{W} - (\sigma_1(\boldsymbol{W}) - \sigma_*)\,\boldsymbol{u}_1(\boldsymbol{W})\boldsymbol{v}_1(\boldsymbol{W})^{\mathsf{T}} \tag{5}$$

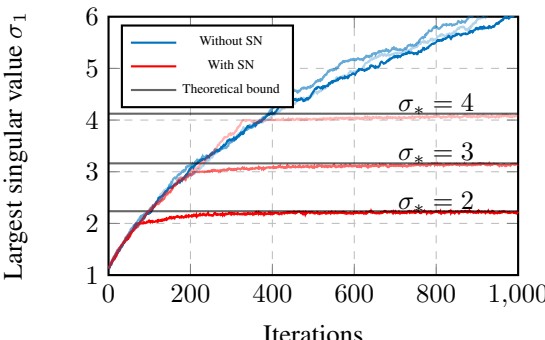

Figure 2: Behavior of the largest singular value of a weight matrix in terms of the iterations of a random walk (see proposition 3.1), without spectral normalization in (blue) and with spectral normalization in (red). The **(black)** lines correspond to the theoretical bound $\sqrt{\sigma_*^2 + \eta^2 d_1 d_0}$ for different $\sigma_*$'s. We took $d_0 = d_1 = 100$ and $\eta = 1/d_0$.

where $\boldsymbol{u}_1(\boldsymbol{W})$ and $\boldsymbol{v}_1(\boldsymbol{W})$ denote respectively the left and right largest singular vectors of $\boldsymbol{W}$, and $\sigma_*$ is an hyper-parameter fixed during training.

To get an insight about the influence of this operation and to ensure that it controls the Lipschitz constant of the generator, the following proposition provides the dynamics of a random walk in the space of parameters along with the spectral normalization in Equation (5). Indeed, since stochastic gradient descent (SGD) consists in estimating the gradient of the loss function on randomly selected batches of data, it can be assimilated to a random walk in the space of parameters (Antognini & Sohl-Dickstein, 2018).

**Proposition 3.1** (Lipschitz constant control). *Let $\sigma_* > 0$ and $\mathcal{G}$ be a neural network composed of $N$ affine layers, each one of input dimension $d_{i-1}$ and output dimension $d_i$ for $i \in [N]$, with $1$-Lipschitz activation functions. Assume that the weights of $\mathcal{G}$ at layer $i + 1$ are initialized as $\mathcal{U}([-\frac{1}{\sqrt{d_i}}, \frac{1}{\sqrt{d_i}}])$, and consider the following dynamics with learning rate $\eta$:*

$$
\begin{aligned}
\boldsymbol{W} &\leftarrow \boldsymbol{W} - \eta \boldsymbol{E}, \text{ with } \boldsymbol{E}_{i,j} \sim \mathcal{N}(0,1) \\
\boldsymbol{W} &\leftarrow \boldsymbol{W} - \max(0, \sigma_1(\boldsymbol{W}) - \sigma_*) \, \boldsymbol{u}_1(\boldsymbol{W}) \boldsymbol{v}_1(\boldsymbol{W})^\mathsf{T}.
\end{aligned}
\tag{6}
$$

*Then, $\forall \varepsilon > 0$, the Lipschitz constant of $\mathcal{G}$ is bounded at convergence with high probability as:*

$$
\|\mathcal{G}\|_{lip} \leq \prod_{i=1}^{N} \left( \varepsilon + \sqrt{\sigma_*^2 + \eta^2 d_i d_{i-1}} \right).
\tag{7}
$$

*Proof.* The proof is provided in Appendix B. □

Proposition 3.1 shows that the Lipschitz constant of a neural network is controlled when trained with the spectral normalization in Equation (5). In particular, recalling the notations in Proposition 3.1, in the limit where $d_i \to \infty$ with $\frac{d_i}{d_{i-1}} \to \gamma_i \in (0, \infty)$ for all $i \in [N]$ and choosing the learning rate $\eta = \mathcal{O}(d_0^{-1})$, the Lipschitz constant of $\mathcal{G}$ is of order $\mathcal{O}(1)$ if it has finitely many layers $N$ and $\sigma_*$ is constant. Therefore, with this spectral normalization, it can be assumed that $\|\mathcal{G}\|_{lip} = \mathcal{O}(1)$ when dimensions grow. Figure 2 depicts the behavior of the Lipschitz constant of a linear layer with and without spectral normalization in the setting of Proposition 3.1, which confirms the obtained bound.

### 3.2 MIXTURE OF CONCENTRATED VECTORS

In this section, we assume data to be a mixture of concentrated random vectors with controlled $\mathcal{O}(1)$ Lipschitz constant (*e.g.*, DL representations of GAN-data as we discussed in the previous section). Precisely, let $\boldsymbol{x}_1, \ldots, \boldsymbol{x}_n$ be a set of mutually independent random vectors in $\mathbb{R}^p$. We suppose that these vectors are distributed as one of $k$ classes of distribution laws $\mu_1, \ldots, \mu_k$ with distinct means $\{\boldsymbol{m}_\ell\}_{\ell=1}^k$ and "covariances" $\{\boldsymbol{C}_\ell\}_{\ell=1}^k$ defined receptively as

$$
\boldsymbol{m}_\ell = \mathbb{E}_{\boldsymbol{x}_i \sim \mu_\ell}[\boldsymbol{x}_i], \quad \boldsymbol{C}_\ell = \mathbb{E}_{\boldsymbol{x}_i \sim \mu_\ell}[\boldsymbol{x}_i \boldsymbol{x}_i^\mathsf{T}].
\tag{8}
$$

For some $q > 0$, we consider a $q$-exponential concentration property on the laws $\mu_\ell$, in the sense that for any family of independent vectors $\boldsymbol{y}_1, \ldots, \boldsymbol{y}_s$ sampled from $\mu_\ell$, $[\boldsymbol{y}_1, \ldots, \boldsymbol{y}_s] \in \mathcal{E}_q(1 \,|\, \mathbb{R}^{p \times s}, \| \cdot \|_F)$. Without loss of generality, we arrange the $\boldsymbol{x}_i$'s in a data matrix $\boldsymbol{X} = [\boldsymbol{x}_1, \ldots, \boldsymbol{x}_n]$ such that, for each $\ell \in [k]$, $\boldsymbol{x}_{1 + \sum_{j=1}^{\ell-1} n_j}, \ldots, \boldsymbol{x}_{\sum_{j=1}^{\ell} n_j} \sim \mu_\ell$, where $n_\ell$ stands for the number of $\boldsymbol{x}_i$'s sampled from $\mu_\ell$. In particular, we have the concentration of $\boldsymbol{X}$ as

$$\boldsymbol{X} \in \mathcal{E}_q(1 \,|\, \mathbb{R}^{p \times n}, \| \cdot \|_F). \tag{9}$$

Such a data matrix $\boldsymbol{X}$ can be constructed through Lipschitz-ally transformed Gaussian vectors ($q = 2$), with controlled Lipschitz constant, thanks to Corollary 2.4. In particular, DL representations of GAN-data are constructed as such, as shown in Section 3.1. We further introduce the following notations that will be used subsequently.

$$\boldsymbol{M} = [\boldsymbol{m}_1, \ldots, \boldsymbol{m}_k] \in \mathbb{R}^{p \times k}, \; \boldsymbol{J} = [\boldsymbol{j}_1, \ldots, \boldsymbol{j}_k] \in \mathbb{R}^{n \times k} \text{ and } \boldsymbol{Z} = [\boldsymbol{z}_1, \ldots, \boldsymbol{z}_n] \in \mathbb{R}^{p \times n},$$

where $\boldsymbol{j}_\ell \in \mathbb{R}^n$ stands for the canonical vector selecting the $\boldsymbol{x}_i$'s of distribution $\mu_\ell$, defined by $(\boldsymbol{j}_\ell)_i = \mathbf{1}_{\boldsymbol{x}_i \sim \mu_\ell}$, and the $\boldsymbol{z}_i$'s are the centered versions of the $\boldsymbol{x}_i$'s, *i.e.* $\boldsymbol{z}_i = \boldsymbol{x}_i - \boldsymbol{m}_\ell$ for $\boldsymbol{x}_i \sim \mu_\ell$.

## 3.3 GRAM MATRICES OF CONCENTRATED VECTORS

Now we study the behavior of the Gram matrix $\boldsymbol{G} = \frac{1}{p} \boldsymbol{X}^\intercal \boldsymbol{X}$ in the large $n, p$ limit and under the model of the previous section. Indeed, $\boldsymbol{G}$ appears as a central component in many classification, regression and clustering methods. Precisely, a finer description of the behavior of $\boldsymbol{G}$ provides access to the internal functioning and performance evaluation of a wide range of machine learning methods such as Least Squares SVMs (AK et al., 2002), Semi-supervised Learning (Chapelle et al., 2009) and Spectral Clustering (Ng et al., 2002). Indeed, the performance evaluation of these methods has already been studied under GMM models in (Liao & Couillet, 2017; Mai & Couillet, 2017; Couillet & Benaych-Georges, 2016) through RMT. On the other hand, analyzing the spectral behavior of $\boldsymbol{G}$ for DL representations quantifies their quality –through its principal subspace (Montavon et al., 2011)– as we have discussed in the introduction. In particular, the Gram matrix decomposes as

$$\boldsymbol{G} = \frac{1}{p} \boldsymbol{J} \boldsymbol{M}^\intercal \boldsymbol{M} \boldsymbol{J}^\intercal + \frac{1}{p} \boldsymbol{Z}^\intercal \boldsymbol{Z} + \frac{1}{p} (\boldsymbol{J} \boldsymbol{M}^\intercal \boldsymbol{Z} + \boldsymbol{Z}^\intercal \boldsymbol{M} \boldsymbol{J}^\intercal). \tag{10}$$

Intuitively $\boldsymbol{G}$ decomposes as a low-rank informative matrix containing the class canonical vectors through $\boldsymbol{J}$ and a noise term represented by the other matrices and essentially $\boldsymbol{Z}^\intercal \boldsymbol{Z}$. Given the form of this decomposition, RMT predicts –through an analysis of the spectrum of $\boldsymbol{G}$ and under a GMM model (Benaych-Georges & Couillet, 2016)– the existence of a threshold $\xi$ function of the ratio $p/n$ and the data statistics for which the dominant eigenvectors of $\boldsymbol{G}$ contain information about the classes only when $\|\boldsymbol{M}^\intercal \boldsymbol{M}\| \geq \xi$ asymptotically (*i.e.*, only when the means of the different classes are sufficiently distinct).

In order to characterize the spectral behavior (*i.e.*, eigenvalues and leading eigenvectors) of $\boldsymbol{G}$ under the concentration assumption in Equation (9) on $\boldsymbol{X}$, we will be interested in determining the spectral distribution $L = \frac{1}{n} \sum_{i=1}^{n} \delta_{\lambda_i}$ of $\boldsymbol{G}$, with $\lambda_1, \ldots, \lambda_n$ the eigenvalues of $\boldsymbol{G}$, where $\delta_x$ stands for the Dirac measure at point $x$. Essentially, to determine the limiting eigenvalue distribution as $p, n \to \infty$ and $p/n \to c \in (0, \infty)$, a conventional approach in RMT consists in determining an estimate of the *Stieltjes transform* (Silverstein & Choi, 1995) $m_L$ of $L$, which is defined for some $z \in \mathbb{C} \setminus \mathrm{Supp}(L)$

$$m_L(z) = \int_\lambda \frac{dL(\lambda)}{\lambda - z} = \frac{1}{n} \mathrm{tr}\left( (\boldsymbol{G} - z \boldsymbol{I}_n)^{-1} \right). \tag{11}$$

Hence, quantifying the behavior of the *resolvent* of $\boldsymbol{G}$ defined as $\boldsymbol{R}(z) = (\boldsymbol{G} + z \boldsymbol{I}_n)^{-1}$ determines the limiting measure of $L$ through $m_L(z)$. Furthermore, since $\boldsymbol{R}(z)$ and $\boldsymbol{G}$ share the same eigenvectors with associated eigenvalues $\frac{1}{\lambda_i - z}$, the projector matrix corresponding to the top $m$ eigenvectors $\boldsymbol{U} = [\boldsymbol{u}_1, \ldots, \boldsymbol{u}_m]$ of $\boldsymbol{G}$ can be calculated through a Cauchy integral $\boldsymbol{U} \boldsymbol{U}^\intercal = \frac{1}{2\pi i} \oint_\gamma \boldsymbol{R}(-z) dz$ where $\gamma$ is an oriented complex contour surrounding the top $m$ eigenvalues of $\boldsymbol{G}$.

To study the behavior of $\boldsymbol{R}(z)$, we look for a so-called *deterministic equivalent* (Hachem et al., 2007) $\tilde{\boldsymbol{R}}(z)$ for $\boldsymbol{R}(z)$, which is a deterministic matrix that satisfies for all $\boldsymbol{A} \in \mathbb{R}^{n \times n}$ and all $\boldsymbol{u}, \boldsymbol{v} \in \mathbb{R}^n$ of respectively bounded spectral and Eucildean norms, $\frac{1}{n} \mathrm{tr}(\boldsymbol{A} \boldsymbol{R}(z)) - \frac{1}{n} \mathrm{tr}(\boldsymbol{A} \tilde{\boldsymbol{R}}(z)) \to 0$ and $\boldsymbol{u}^\intercal (\boldsymbol{R}(z) - \tilde{\boldsymbol{R}}(z)) \boldsymbol{v} \to 0$ almost surely as $n \to \infty$. In the following, we present our main result which gives such a deterministic equivalent under the concentration assumption on $\boldsymbol{X}$ in Equation (9) and under the following assumptions.

**Assumption 3.2.** *As $p \to \infty$,*

*1. $p/n \to c \in (0, \infty)$,      2. The number of classes $k$ is bounded,      3. $\|\boldsymbol{m}_\ell\| = \mathcal{O}(\sqrt{p})$.*

**Theorem 3.3** (Deterministic Equivalent for $\boldsymbol{R}(z)$). *Under the model described in Section 3.2 and Assumptions 3.2, we have $\boldsymbol{R}(z) \in \mathcal{E}_q(p^{-1/2} \,|\, \mathbb{R}^{n \times n}, \|\cdot\|_F)$. Furthermore,*

$$\left\| \mathbb{E}\boldsymbol{R}(z) - \tilde{\boldsymbol{R}}(z) \right\| = \mathcal{O}\left( \sqrt{\frac{\log(p)}{p}} \right) , \; \tilde{\boldsymbol{R}}(z) = \frac{1}{z} \operatorname{diag}\left\{ \frac{\boldsymbol{I}_{n_\ell}}{1 + \delta_\ell^*(z)} \right\}_{\ell=1}^k + \frac{1}{p\,z} \boldsymbol{J}\boldsymbol{\Omega}_z \boldsymbol{J}^\mathsf{T} \quad (12)$$

*with $\boldsymbol{\Omega}_z = \boldsymbol{M}^\mathsf{T} \tilde{\boldsymbol{Q}}(z) \boldsymbol{M} \odot \operatorname{diag}\left\{ \frac{\delta_\ell^*(z)-1}{\delta_\ell^*(z)+1} \right\}_{\ell=1}^k$ and $\tilde{\boldsymbol{Q}}(z) = \left( \frac{1}{c\,k} \sum_{\ell=1}^k \frac{\boldsymbol{C}_\ell}{1+\delta_\ell^*(z)} + z\boldsymbol{I}_p \right)^{-1}$,*

*where $\delta^*(z) = [\delta_1^*(z), \dots, \delta_k^*(z)]^\mathsf{T}$ is the unique fixed point of the system of equations*

$$\delta_\ell(z) = \frac{1}{p} \operatorname{tr}\left( \boldsymbol{C}_\ell \left( \frac{1}{c\,k} \sum_{j=1}^k \frac{\boldsymbol{C}_j}{1+\delta_j(z)} + z\boldsymbol{I}_p \right)^{-1} \right) \; \text{for each } \ell \in [k].$$

*Sketch of proof.* The first step of the proof is to show the concentration of $\boldsymbol{R}(z)$. This comes from the fact that the application $\boldsymbol{X} \mapsto \boldsymbol{R}(z)$ is $2z^{-3/2}p^{-1/2}$-Lipschitz w.r.t. the Frobenius norm, thus we have by Proposition 2.3 that $\boldsymbol{R}(z) \in \mathcal{E}_q(p^{-1/2} \,|\, \mathbb{R}^{n \times n}, \|\cdot\|_F)$. The second step consists in estimating $\mathbb{E}\boldsymbol{R}(z)$ through a deterministic matrix $\tilde{\boldsymbol{R}}(z)$. Indeed, $\boldsymbol{R}(z)$ can be expressed as a function of $\boldsymbol{Q}(z) = (\boldsymbol{X}\boldsymbol{X}^\mathsf{T}/p + z\boldsymbol{I}_p)^{-1}$ as $\boldsymbol{R}(z) = z^{-1}(\boldsymbol{I}_n - \boldsymbol{X}^\mathsf{T}\boldsymbol{Q}(z)\boldsymbol{X}/p)$, and exploiting the result of (Louart & Couillet, 2019) which shows that $\mathbb{E}\boldsymbol{Q}(z)$ can be estimated through $\tilde{\boldsymbol{Q}}(z)$, we obtain the estimator $\tilde{\boldsymbol{R}}(z)$ for $\mathbb{E}\boldsymbol{R}(z)$. A more detailed proof is provided in Section A.3 of the Appendix. □

This result allows specifically to (i) describe the limiting eigenvalues distribution of $\boldsymbol{G}$, (ii) determine the spectral detectability threshold mentioned above, (iii) evaluate the asymptotic "content" of the leading eigenvectors of $\boldsymbol{G}$ and, much more fundamentally, (iv) infer the asymptotic performances of machine learning algorithms that are based on simple functionals of $\boldsymbol{G}$ (*e.g.*, LS-SVM, spectral clustering etc.). Looking carefully at Theorem 3.3 we see that the spectral behavior of the Gram matrix $\boldsymbol{G}$ computed on concentrated vectors only depends on the *first* and *second* order statistics of the laws $\mu_\ell$ (their means $\boldsymbol{m}_\ell$ and "covariances" $\boldsymbol{C}_\ell$). This suggests the surprising result that $\boldsymbol{G}$ has the same behavior as when the data follow a GMM model with the same means and covariances. The asymptotic spectral behavior of $\boldsymbol{G}$ is therefore *universal* with respect to the data distribution laws which satisfy the aforementioned concentration properties (for instance DL representations of GAN-data). We illustrate this universality result in the next section by considering data as CNN representations of GAN generated images.

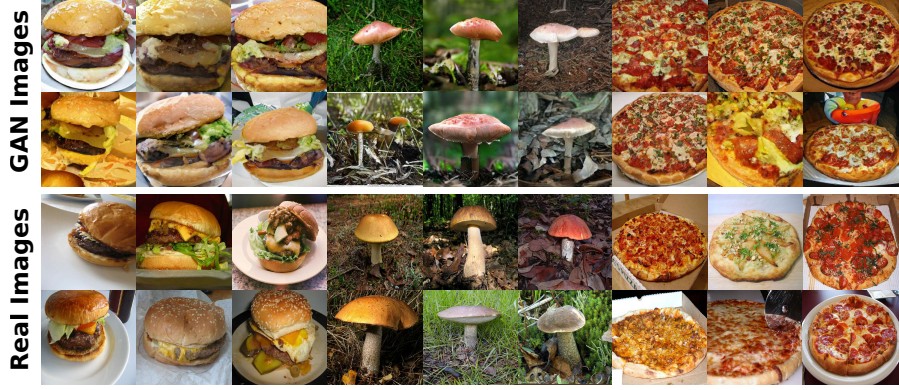

Figure 3: **(Top)** GAN generated images using the BigGAN model Brock et al. (2018). **(Bottom)** Real images selected from the Imagenet dataset Deng et al. (2009). We considered $n = 1500$ images from $k = 3$ classes which are Mushroom, Pizza and Hamburger.

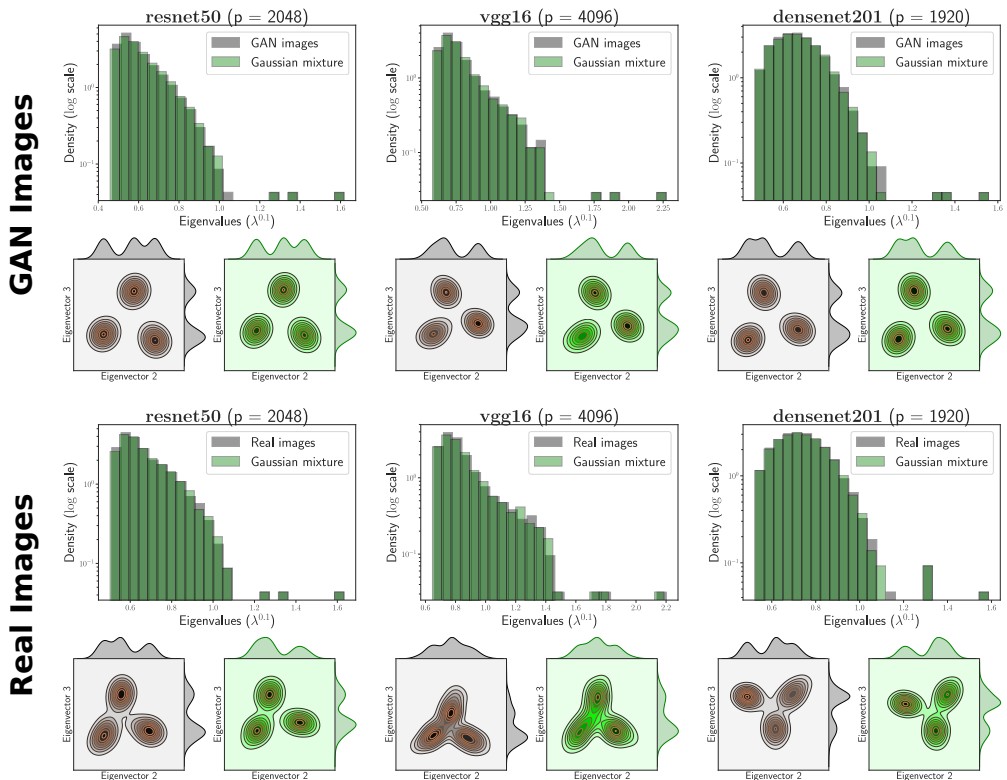

Figure 4: **(Top)** Spectrum and leading eigenspace of the Gram matrix for CNN representations of GAN generated images using the BigGAN model Brock et al. (2018). **(Bottom)** Spectrum and leading eigenspace of the Gram matrix for CNN representations of real images selected from the Imagenet dataset Deng et al. (2009). Columns correspond to the three representation networks (Resnet50, VGG16 and Densenet201).

# 4 APPLICATION TO CNN REPRESENTATIONS OF GAN-GENERATED IMAGES

In this section, we consider $n = 1500$ data $\boldsymbol{x}_1, \ldots, \boldsymbol{x}_n \in \mathbb{R}^p$ as CNN representations –across popular CNN architectures of different sizes $p$– of GAN-generated images using the generator of the Big-GAN model (Brock et al., 2018). We further use real images from the Imagenet dataset (Deng et al., 2009) for comparison. In particular, we empirically compare the spectrum of the Gram matrix of this data with the Gram matrix of a GMM model with the same means and covariances. We also consider the leading 2-dimensional eigenspace of the Gram matrix which contains clustering information as detailed in the previous section. Figure 3 depicts some images generated using the Big-GAN model (Top) and the corresponding real class images from the Imagenet dataset (Bottom). The Big-GAN model is visually able to generate highly realistic images which are by construction concentrated vectors, as discussed in Section 3.1.

Figure 4 depicts the spectrum and leading 2D eigenspace of the Gram matrix computed on CNN representations of GAN generated and real images (in gray), and the corresponding GMM model with same first and second order statistics (in green). The Gram matrix is seen to follow the same spectral behavior for GAN-data as for the GMM model which is a natural consequence of the universality result of Theorem 3.3 with respect to the data distribution. Besides, and perhaps no longer surprisingly, we further observe that the spectral properties of $\boldsymbol{G}$ for real data (here CNN representations of real images) are conclusively matched by their Gaussian counterpart. This both theoretically and empirically confirms that the proposed random matrix framework is fully compliant with the theoretical analysis of real machine learning datasets.

## 5  CONCLUSION

Leveraging on random matrix theory (RMT) and the concentration of measure phenomenon, we have shown through this paper that DL representations of GAN-data behave as Gaussian mixtures for linear classifiers, a fundamental *universal* property which is only valid in high-dimension of data. To the best of our knowledge, this result constitutes a new approach towards the theoretical understanding of complex objects such as DL representations, as well as the understanding of the behavior of more elaborate machine learning algorithms for complex data structures. In addition, the article explicitly demonstrated our ability, through RMT, to anticipate the behavior of a wide range of standard classifiers for data as complex as DL representations of the realistic and surprising images generated by GANs. This opens the way to a more systematic analysis and improvement of machine learning algorithms on real datasets by means of large dimensional statistics.

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

## A  PROOF OF THEOREM 3.3

### A.1  SETTING OF THE PROOF

For simplicity, we will only suppose the case $k = 1$ and we consider the following notations that will be used subsequently.

$$\bar{\boldsymbol{x}} = \mathbb{E}\boldsymbol{x}_i, \ \boldsymbol{C} = \mathbb{E}[\boldsymbol{x}_i\boldsymbol{x}_i^\intercal], \ \boldsymbol{X}_0 = \boldsymbol{X} - \bar{\boldsymbol{x}}\mathbf{1}_n^\intercal, \ \boldsymbol{C}_0 = \mathbb{E}[\boldsymbol{X}_0\boldsymbol{X}_0^\intercal/n].$$

Let

$$\boldsymbol{X}_{-i} = (\boldsymbol{x}_1, \ldots, \boldsymbol{x}_{i-1}, 0, \boldsymbol{x}_i, \ldots, \boldsymbol{x}_n)$$

the matrix $X$ with a vector of zeros at its $i$th column.

Denote the resolvents

$$\boldsymbol{R} = \left(\frac{\boldsymbol{X}^\intercal \boldsymbol{X}}{p} + z\boldsymbol{I}_n\right)^{-1}, \ \boldsymbol{Q} = \left(\frac{\boldsymbol{X}\boldsymbol{X}^\intercal}{p} + z\boldsymbol{I}_p\right)^{-1}, \ \boldsymbol{Q}_{-i} = \left(\frac{\boldsymbol{X}\boldsymbol{X}^\intercal}{p} - \frac{\boldsymbol{x}_i\boldsymbol{x}_i^\intercal}{p} + z\boldsymbol{I}_p\right)^{-1} \tag{13}$$

And let

$$\tilde{\boldsymbol{Q}} = \left(\frac{1}{c}\frac{\boldsymbol{C}}{1+\delta} + z\boldsymbol{I}_p\right)^{-1}, \tag{14}$$

where $\delta$ is the solution to the fixed point equation

$$\delta = \frac{1}{p} \operatorname{tr} \left( C \left( \frac{1}{c} \frac{C}{1+\delta} + z I_p \right)^{-1} \right).$$

## A.2 BASIC TOOLS

**Lemma A.1** ((Ledoux, 2005)). *Let $z \in \mathcal{E}_q(1 \mid \mathbb{R}^p, \| \cdot \|)$ and $M \in \mathcal{E}_q(1 \mid \mathbb{R}^{p \times n}, \| \cdot \|_F)$. Then, for some numerical constant $C > 0$*

- $\mathbb{E} \|z\| \leq \|\mathbb{E}z\| + C\sqrt{p}, \ \ \mathbb{E} \|z\|_\infty \leq \|\mathbb{E}z\|_\infty + C\sqrt{\log p}.$
- $\mathbb{E} \|M\| \leq \|\mathbb{E}M\| + C\sqrt{p+n}, \ \ \mathbb{E} \|M\|_F \leq \|\mathbb{E}M\|_F + C\sqrt{pn}.$

**Lemma A.2.** *Denote $Q_{\bar{x}} = (\bar{x}\bar{x}^\mathsf{T} + z I_p)^{-1}$, we have:*

$$Q_{\bar{x}}\bar{x} = \frac{\bar{x}}{\|\bar{x}\|^2 + z} \ \ \text{and} \ \ \|\tilde{Q}\bar{x}\|, \ \bar{x}\tilde{Q}\bar{x} = \mathcal{O}(1).$$

*Moreover, if $\|\bar{x}\| \geq \sqrt{p}$, $\|\tilde{Q}\bar{x}\| = \mathcal{O}(p^{-1/2})$.*

*Proof.* Since $zQ_{\bar{x}} = I_p - Q_{\bar{x}}\bar{x}\bar{x}^\mathsf{T}$ :

$$zQ_{\bar{x}}\bar{x} = \bar{x} - \|\bar{x}\|^2 Q_{\bar{x}}\bar{x},$$

and we recover the first identity of the Lemma.

And since the matrix $C_0$ is nonnegative symmetric, we have :

$$\tilde{Q}\bar{x} = \left( \frac{1}{c} \frac{C_0 + \bar{x}\bar{x}^\mathsf{T}}{1+\delta} + z I_p \right)^{-1} \bar{x} \leq \frac{c(1+\delta)\bar{x}}{\|\bar{x}\|^2 + zc(1+\delta)}.$$

Therefore, $\bar{x}\tilde{Q}\bar{x} = \frac{c(1+\delta)\|\bar{x}\|^2}{\|\bar{x}\|^2 + zc(1+\delta)} = \mathcal{O}(1)$ and:

$$\|\tilde{Q}\bar{x}\| = \frac{c(1+\delta)\|\bar{x}\|}{\|\bar{x}\|^2 + zc(1+\delta)} \leq \begin{cases} \dfrac{\|\bar{x}\|}{z} = \mathcal{O}(1) \ \text{if} \ \|\bar{x}\| \leq 1, \\[2ex] \dfrac{c(1+\delta)}{\|\bar{x}\|} = \mathcal{O}(1) \ \text{if} \ \|\bar{x}\| \geq 1. \end{cases}$$

$\square$

**Proposition A.3.** $\bar{x}^\mathsf{T} \mathbb{E}[Q]\bar{x} = \bar{x}^\mathsf{T} \tilde{Q}\bar{x} + \mathcal{O}\left( \sqrt{\frac{\log p}{p}} \right)$

*Proof.* Let us bound:

$$\left| \bar{x}^\mathsf{T} Q\bar{x} - \bar{x}^\mathsf{T} \tilde{Q}\bar{x} \right| \leq \frac{c^{-1}}{1+\delta} \left| \mathbb{E} \left[ \bar{x}Q x_i x_i^\mathsf{T} \tilde{Q}\bar{x} \left( \frac{1}{p} x_i^\mathsf{T} Q_{-i} x_i - \delta \right) \right] + \frac{1}{p} \mathbb{E} \left[ \bar{x}^\mathsf{T} Q_{-i} x_i x_i^\mathsf{T} Q C \tilde{Q}\bar{x} \right] \right|$$

Now let us consider a supplementary random vector $x_{n+1}$ following the same low as the $x_i$'s and independent of $X$. We divide the set $\mathbb{I} = [n+1]$ into two sets $\mathbb{I}_{\frac{1}{2}}$ and $\mathbb{I}_{\frac{2}{2}}$ of same cardinality ($\lfloor \frac{n+1}{2} \rfloor \leq \#\mathbb{I}_{\frac{1}{2}}, \#\mathbb{I}_{\frac{2}{2}} \leq \lceil \frac{n+1}{2} \rceil$), we note $X_{\frac{1}{2}} = (x_i \mid i \in \mathbb{I}_{\frac{1}{2}})$, $X_{\frac{2}{2}} = (x_i \mid i \in \mathbb{I}_{\frac{2}{2}})$ and we introduce the diagonal matrices $\Delta = \operatorname{diag}\left( \frac{1}{p} x_i^\mathsf{T} Q_{-i} x_i - \delta \mid i \in \mathbb{I}_{\frac{1}{2}} \right)$, $D = \operatorname{diag}\left( 1 + \frac{1}{p+1} x_i^\mathsf{T} Q x_i \mid i \in \mathbb{I}_{\frac{2}{2}} \right)$.

We have the bound:

$$\left| \mathbb{E}\left[ \bar{\boldsymbol{x}} \boldsymbol{Q} \boldsymbol{x}_i \boldsymbol{x}_i^{\mathsf{T}} \tilde{\boldsymbol{Q}} \bar{\boldsymbol{x}} \left( \frac{1}{p} \boldsymbol{x}_i^{\mathsf{T}} \boldsymbol{Q}_{-i} \boldsymbol{x}_i - \delta \right) \right] \right|$$

$$= \left| \mathbb{E}\left[ \left( 1 + \frac{1}{p} \boldsymbol{x}_{n+1}^{\mathsf{T}} \boldsymbol{Q} \boldsymbol{x}_{n+1} \right) \boldsymbol{x}_{n+1} \boldsymbol{Q}_{+(n+1)} \boldsymbol{x}_i \boldsymbol{x}_i^{\mathsf{T}} \tilde{\boldsymbol{Q}} \bar{\boldsymbol{x}} \left( \frac{1}{p} \boldsymbol{x}_i^{\mathsf{T}} \boldsymbol{Q}_{-i} \boldsymbol{x}_i - \delta \right) \right] \right|$$

$$= \frac{1}{p^2} \left| \mathbb{E}\left[ \boldsymbol{1}^{\mathsf{T}} \boldsymbol{D} \boldsymbol{X}_{\frac{1}{2}}^{\mathsf{T}} \boldsymbol{Q}_{+(n+1)} \boldsymbol{X}_{\frac{1}{2}} \boldsymbol{\Delta} \boldsymbol{X}_{\frac{1}{2}}^{\mathsf{T}} \tilde{\boldsymbol{Q}} \bar{\boldsymbol{x}} \right] \right|$$

$$\leq \sqrt{ \left| \mathbb{E}\left[ \frac{1}{p^3} \boldsymbol{1}^{\mathsf{T}} \boldsymbol{D} \boldsymbol{X}_{\frac{1}{2}}^{\mathsf{T}} \boldsymbol{Q}_{+(n+1)} \boldsymbol{X}_{\frac{1}{2}} \boldsymbol{\Delta}^2 \boldsymbol{X}_{\frac{1}{2}}^{\mathsf{T}} \boldsymbol{Q}_{+(n+1)} \boldsymbol{X}_{\frac{1}{2}} \boldsymbol{D} \boldsymbol{1} \right] \mathbb{E}\left[ \frac{1}{p} \bar{\boldsymbol{x}}^{\mathsf{T}} \tilde{\boldsymbol{Q}} \boldsymbol{X}_{\frac{1}{2}} \boldsymbol{X}_{\frac{1}{2}}^{\mathsf{T}} \tilde{\boldsymbol{Q}} \bar{\boldsymbol{x}} \right] \right| }$$

$$\leq \sqrt{ \left| \mathbb{E}\left[ \left\| \frac{1}{p} \boldsymbol{X}_{\frac{1}{2}}^{\mathsf{T}} \boldsymbol{Q}_{+(n+1)} \boldsymbol{X}_{\frac{1}{2}} \right\|^2 \|\boldsymbol{D}\|^2 \|\boldsymbol{\Delta}\|^2 \right] \mathbb{E}\left[ \bar{\boldsymbol{x}} \tilde{\boldsymbol{Q}} \boldsymbol{C} \tilde{\boldsymbol{Q}} \bar{\boldsymbol{x}} \right] \right| } \leq \mathcal{O}\left( \sqrt{ \frac{\log p}{p} } \right),$$

thanks to Lemma A.1 and Lemma A.2 (the spectral norm of $\boldsymbol{\Delta}$ and $\boldsymbol{D}$ is just an infinity norm if we see them as random vectors of $\mathbb{R}^n$). We can bound $\frac{1}{p} \left| \mathbb{E}\left[ \bar{\boldsymbol{x}}^{\mathsf{T}} \boldsymbol{Q}_{-i} \boldsymbol{x}_i \boldsymbol{x}_i^{\mathsf{T}} \boldsymbol{Q} \boldsymbol{C} \tilde{\boldsymbol{Q}} \bar{\boldsymbol{x}} \right] \right|$ the same way to obtain the result of the proposition. $\qquad\square$

**Proposition A.4.** $\|\mathbb{E}[\boldsymbol{x}_i^{\mathsf{T}} \boldsymbol{Q}_{-i} \boldsymbol{X}_{-i}] - \frac{\bar{\boldsymbol{x}}^{\mathsf{T}} \tilde{\boldsymbol{Q}} \bar{\boldsymbol{x}} \boldsymbol{1}^{\mathsf{T}}}{1+\delta}\| = \mathcal{O}(\sqrt{\log p})$

*Proof.* Considering $\boldsymbol{u} \in \mathbb{R}^n$ such that $\|\boldsymbol{u}\| = 1$:

$$\left| \mathbb{E}[\boldsymbol{x}_i^{\mathsf{T}} \boldsymbol{Q}_{-i} \boldsymbol{X}_{-i} \boldsymbol{u}] - \frac{\bar{\boldsymbol{x}}^{\mathsf{T}} \tilde{\boldsymbol{Q}} \bar{\boldsymbol{x}} \boldsymbol{1}^{\mathsf{T}} \boldsymbol{u}}{1+\delta} \right|$$

$$= \left| \sum_{\substack{j=1 \\ j \neq i}}^n \boldsymbol{u}_j \mathbb{E}\left[ \frac{\boldsymbol{x}_i^{\mathsf{T}} \boldsymbol{Q}_{-j}^{-i} \boldsymbol{x}_j}{1 + \frac{1}{p} \boldsymbol{x}_j^{\mathsf{T}} \boldsymbol{Q}_{-i}^{-j} \boldsymbol{x}_j} - \frac{\boldsymbol{x}_i^{\mathsf{T}} \tilde{\boldsymbol{Q}} \boldsymbol{x}_j}{1+\delta} \right] \right|$$

$$\leq \sqrt{n} \left| \mathbb{E}\left[ \frac{\boldsymbol{x}_i^{\mathsf{T}} \boldsymbol{Q}_{-j}^{-i} \boldsymbol{x}_j}{1 + \frac{1}{p} \boldsymbol{x}_j^{\mathsf{T}} \boldsymbol{Q}_{-i}^{-j} \boldsymbol{x}_j} - \frac{\boldsymbol{x}_i^{\mathsf{T}} \boldsymbol{Q}_{-j}^{-i} \boldsymbol{x}_j}{1+\delta} \right] \right| + \left| \frac{1}{1+\delta} \mathbb{E}\left[ \boldsymbol{x}_i^{\mathsf{T}} \boldsymbol{Q}_{-j}^{-i} \boldsymbol{x}_j - \boldsymbol{x}_i^{\mathsf{T}} \tilde{\boldsymbol{Q}} \boldsymbol{x}_j \right] \right| \quad \text{(where } i \neq j\text{)}$$

$$\leq \sqrt{n} \left| \mathbb{E}\left[ \bar{\boldsymbol{x}}^{\mathsf{T}} \boldsymbol{Q} \boldsymbol{x}_j \left( \frac{1}{p} \boldsymbol{x}_j^{\mathsf{T}} \boldsymbol{Q}_{-i}^{-j} \boldsymbol{x}_j - \delta \right) \right] \right| + \sqrt{n} \left| \mathbb{E}\left[ \bar{\boldsymbol{x}}^{\mathsf{T}} \boldsymbol{Q}_{-j}^{-i} \bar{\boldsymbol{x}} - \bar{\boldsymbol{x}}^{\mathsf{T}} \tilde{\boldsymbol{Q}} \bar{\boldsymbol{x}} \right] \right|,$$

where the first term is treated the same way as we did in the proof of Proposition A.3 and the second term is bounded thanks to Proposition A.3 $\qquad\square$

### A.3 MAIN BODY OF THE PROOF

*Proof of Theorem 3.3.* Recall the definition of the resolvents $\boldsymbol{R}$ and $\boldsymbol{Q}$ in Equation (13). The first step of the proof is to show the concentration of $\boldsymbol{R}$. This comes from the fact that the application $\Phi : \boldsymbol{X} \mapsto (\boldsymbol{X}^{\mathsf{T}} \boldsymbol{X} + z \boldsymbol{I}_n)^{-1}$ is $2z^{-3/2}$-Lipschitz *w.r.t.* the Frobenius norm. Indeed, by the matrix identity $\boldsymbol{A} - \boldsymbol{B} = \boldsymbol{A}(\boldsymbol{B}^{-1} - \boldsymbol{A}^{-1})\boldsymbol{B}$, we have

$$\Phi(\boldsymbol{X}) - \Phi(\boldsymbol{X} + \boldsymbol{H}) = \Phi(\boldsymbol{X})(\boldsymbol{H}^{\mathsf{T}} \boldsymbol{X} + (\boldsymbol{X} + \boldsymbol{H})^{\mathsf{T}} \boldsymbol{H})\Phi(\boldsymbol{X} + \boldsymbol{H})$$

And by the bounds $\|\boldsymbol{A}\boldsymbol{B}\|_F \leq \|\boldsymbol{A}\| \cdot \|\boldsymbol{B}\|_F$, $\|\Phi(\boldsymbol{X})\boldsymbol{X}^{\mathsf{T}}\| \leq z^{-1/2}$ and $\|\Phi(\boldsymbol{X})\| \leq z^{-1}$, we have

$$\|\Phi(\boldsymbol{X} + \boldsymbol{H}) - \Phi(\boldsymbol{X})\|_F \leq \frac{2}{z^{3/2}} \|\boldsymbol{H}\|_F.$$

Therefore, given $\boldsymbol{X} \in \mathcal{E}_q(1 \,|\, \mathbb{R}^{p \times n}, \|\cdot\|_F)$ and since the application $\boldsymbol{X} \mapsto \boldsymbol{R} = \Phi(\boldsymbol{X}/\sqrt{p})$ is $2z^{-3/2} p^{-1/2}$-Lipschitz, we have by Proposition 2.3 that $\boldsymbol{R} \in \mathcal{E}_q(p^{-1/2} \,|\, \mathbb{R}^{n \times n}, \|\cdot\|_F)$.

The second step consists in estimating $\mathbb{E}\boldsymbol{R}(z)$ through a deterministic matrix $\tilde{\boldsymbol{R}}$. Indeed, by the identity $(\boldsymbol{M}^{\mathsf{T}} \boldsymbol{M} + z \boldsymbol{I})^{-1} \boldsymbol{M}^{\mathsf{T}} = \boldsymbol{M}^{\mathsf{T}} (\boldsymbol{M} \boldsymbol{M}^{\mathsf{T}} + z \boldsymbol{I})^{-1}$, the resolvent $\boldsymbol{R}$ can be expressed in function

of $\boldsymbol{Q}$ as follows

$$\boldsymbol{R} = \frac{1}{z}\left(I_n - \frac{\boldsymbol{X}^\mathsf{T}\boldsymbol{Q}\boldsymbol{X}}{p}\right), \tag{15}$$

thus a deterministic equivalent for $\boldsymbol{R}$ can therefore be obtained through a deterministic equivalent of the matrix $\boldsymbol{X}^\mathsf{T}\boldsymbol{Q}\boldsymbol{X}$. However, as demonstrated in Louart & Couillet (2019), the matrix $\boldsymbol{Q}$ has as a deterministic equivalent the matrix $\tilde{\boldsymbol{Q}}$ defined in equation 14. In the following, we aim at deriving a deterministic equivalent for $\frac{1}{p}\boldsymbol{X}^\mathsf{T}\boldsymbol{Q}\boldsymbol{X}$ in function of $\tilde{\boldsymbol{Q}}$. Let $\boldsymbol{u}$ and $\boldsymbol{v}$ be two unitary vectors in $\mathbb{R}^n$, and let us estimate

$$\Delta \equiv \mathbb{E}\left[\boldsymbol{u}^\mathsf{T}\left(\frac{\boldsymbol{X}^\mathsf{T}\boldsymbol{Q}\boldsymbol{X}}{p} - \frac{\boldsymbol{X}^\mathsf{T}\tilde{\boldsymbol{Q}}\boldsymbol{X}}{p}\right)\boldsymbol{v}\right] = \frac{1}{p}\mathbb{E}\left[\frac{\boldsymbol{u}^\mathsf{T}\boldsymbol{X}^\mathsf{T}\boldsymbol{Q}\boldsymbol{C}\tilde{\boldsymbol{Q}}\boldsymbol{X}\boldsymbol{v}}{1+\delta} - \frac{1}{p}\boldsymbol{u}^\mathsf{T}\boldsymbol{X}^\mathsf{T}\boldsymbol{Q}\boldsymbol{X}\boldsymbol{X}^\mathsf{T}\tilde{\boldsymbol{Q}}\boldsymbol{X}\boldsymbol{v}\right]$$

With the following matrix identities (to explore the independence of the columns of $\boldsymbol{X}$):

$$\boldsymbol{Q} = \boldsymbol{Q}_{-i} - \frac{1}{p}\boldsymbol{Q}_{-i}\boldsymbol{x}_i\boldsymbol{x}_i^\mathsf{T}\boldsymbol{Q}, \qquad \boldsymbol{Q}\boldsymbol{x}_i = \frac{\boldsymbol{Q}_{-i}\boldsymbol{x}_i}{1+\frac{1}{p}\boldsymbol{x}_i^\mathsf{T}\boldsymbol{Q}_{-i}\boldsymbol{x}_i}, \qquad \boldsymbol{A} - \boldsymbol{B} = \boldsymbol{A}(\boldsymbol{B}^{-1} - \boldsymbol{A}^{-1})\boldsymbol{B}$$

and the decomposition $\boldsymbol{Q}\boldsymbol{X}\boldsymbol{X}^\mathsf{T} = \sum_{i=1}^n \boldsymbol{Q}\boldsymbol{x}_i\boldsymbol{x}_i^\mathsf{T}$, we obtain:

$$\Delta = \frac{1}{p^2}\mathbb{E}\left[\sum_{i=1}^n \frac{\boldsymbol{u}^\mathsf{T}\boldsymbol{X}^\mathsf{T}\boldsymbol{Q}_{-i}\boldsymbol{C}\tilde{\boldsymbol{Q}}\boldsymbol{X}\boldsymbol{v}}{1+\delta} - \frac{\boldsymbol{u}^\mathsf{T}\boldsymbol{X}^\mathsf{T}\boldsymbol{Q}_{-i}\boldsymbol{x}_i\boldsymbol{x}_i^\mathsf{T}\tilde{\boldsymbol{Q}}\boldsymbol{X}\boldsymbol{v}}{1+\frac{1}{p}\boldsymbol{x}_i^\mathsf{T}\boldsymbol{Q}_{-i}\boldsymbol{x}_i} - \frac{1}{p}\frac{\boldsymbol{u}^\mathsf{T}\boldsymbol{X}^\mathsf{T}\boldsymbol{Q}_{-i}\boldsymbol{x}_i\boldsymbol{x}_i^\mathsf{T}\boldsymbol{Q}\boldsymbol{C}\tilde{\boldsymbol{Q}}\boldsymbol{X}\boldsymbol{v}}{1+\delta}\right]$$

$$= \frac{1}{p^2}\sum_{i=1}^n \mathbb{E}\left[\frac{\boldsymbol{u}^\mathsf{T}\boldsymbol{X}_{-i}^\mathsf{T}\boldsymbol{Q}_{-i}\boldsymbol{C}\tilde{\boldsymbol{Q}}\boldsymbol{X}_{-i}\boldsymbol{v}}{1+\delta} - \frac{\boldsymbol{u}^\mathsf{T}\boldsymbol{X}_{-i}^\mathsf{T}\boldsymbol{Q}_{-i}\boldsymbol{x}_i\boldsymbol{x}_i^\mathsf{T}\tilde{\boldsymbol{Q}}\boldsymbol{X}_{-i}\boldsymbol{v}}{1+\frac{1}{p}\boldsymbol{x}_i^\mathsf{T}\boldsymbol{Q}_{-i}\boldsymbol{x}_i}\right.$$

$$+ \frac{u_i\boldsymbol{x}_i^\mathsf{T}\boldsymbol{Q}_{-i}\boldsymbol{C}\tilde{\boldsymbol{Q}}\boldsymbol{X}_{-i}\boldsymbol{v}}{1+\delta} + \frac{v_i\boldsymbol{u}^\mathsf{T}\boldsymbol{X}_{-i}^\mathsf{T}\boldsymbol{Q}_{-i}\boldsymbol{C}\tilde{\boldsymbol{Q}}\boldsymbol{x}_i}{1+\delta} + u_iv_i\frac{\boldsymbol{x}_i^\mathsf{T}\boldsymbol{Q}_{-i}\boldsymbol{C}\tilde{\boldsymbol{Q}}\boldsymbol{x}_i}{1+\delta}$$

$$- \frac{u_i\boldsymbol{x}_i^\mathsf{T}\boldsymbol{Q}_{-i}\boldsymbol{x}_i\boldsymbol{x}_i^\mathsf{T}\tilde{\boldsymbol{Q}}\boldsymbol{X}_{-i}\boldsymbol{v}}{1+\frac{1}{p}\boldsymbol{x}_i^\mathsf{T}\boldsymbol{Q}_{-i}\boldsymbol{x}_i} - \frac{v_i\boldsymbol{u}^\mathsf{T}\boldsymbol{X}_{-i}^\mathsf{T}\boldsymbol{Q}_{-i}\boldsymbol{x}_i\boldsymbol{x}_i^\mathsf{T}\tilde{\boldsymbol{Q}}\boldsymbol{x}_i}{1+\frac{1}{p}\boldsymbol{x}_i^\mathsf{T}\boldsymbol{Q}_{-i}\boldsymbol{x}_i} - u_iv_i\frac{\boldsymbol{x}_i^\mathsf{T}\boldsymbol{Q}_{-i}\boldsymbol{x}_i\boldsymbol{x}_i^\mathsf{T}\tilde{\boldsymbol{Q}}\boldsymbol{x}_i}{1+\frac{1}{p}\boldsymbol{x}_i^\mathsf{T}\boldsymbol{Q}_{-i}\boldsymbol{x}_i}$$

$$\left. - \frac{1}{p}\frac{\boldsymbol{u}^\mathsf{T}\boldsymbol{X}^\mathsf{T}\boldsymbol{Q}_{-i}\boldsymbol{x}_i\boldsymbol{x}_i^\mathsf{T}\boldsymbol{Q}\boldsymbol{C}\tilde{\boldsymbol{Q}}\boldsymbol{X}\boldsymbol{v}}{1+\delta}\right]$$

We can show with Holder's inequality and the concentration bounds (mainly the fact that $\frac{1}{p}\boldsymbol{x}_i^\mathsf{T}\boldsymbol{Q}_{-i}\boldsymbol{x}_i$ concentrates around $\delta$) developed in (Louart & Couillet, 2019), that most of the above quantities vanish asymptotically. As a toy example, we consider the following term:

$$\left|\frac{1}{p^2}\sum_{i=1}^n \mathbb{E}\left[\frac{\boldsymbol{u}^\mathsf{T}\boldsymbol{X}_{-i}^\mathsf{T}\boldsymbol{Q}_{-i}\boldsymbol{C}\tilde{\boldsymbol{Q}}\boldsymbol{X}_{-i}\boldsymbol{v}}{1+\delta} - \frac{\boldsymbol{u}^\mathsf{T}\boldsymbol{X}_{-i}^\mathsf{T}\boldsymbol{Q}_{-i}\boldsymbol{x}_i\boldsymbol{x}_i^\mathsf{T}\tilde{\boldsymbol{Q}}\boldsymbol{X}_{-i}\boldsymbol{v}}{1+\frac{1}{p}\boldsymbol{x}_i^\mathsf{T}\boldsymbol{Q}_{-i}\boldsymbol{x}_i}\right]\right|$$

$$= \left|\frac{1}{p^2}\sum_{i=1}^n \mathbb{E}\left[\boldsymbol{u}^\mathsf{T}\boldsymbol{X}_{-i}^\mathsf{T}\boldsymbol{Q}_{-i}\boldsymbol{x}_i\boldsymbol{x}_i^\mathsf{T}\tilde{\boldsymbol{Q}}\boldsymbol{X}_{-i}\boldsymbol{v}\frac{\delta - \frac{1}{p}\boldsymbol{x}_i^\mathsf{T}\boldsymbol{Q}_{-i}\boldsymbol{x}_i}{(1+\delta)(1+\frac{1}{p}\boldsymbol{x}_i^\mathsf{T}\boldsymbol{Q}_{-i}\boldsymbol{x}_i)}\right]\right|$$

$$\leq \left|\frac{1}{p^2}\sum_{i=1}^n \mathbb{E}\left[(\boldsymbol{u}^\mathsf{T}\boldsymbol{X}_{-i}^\mathsf{T}\boldsymbol{Q}_{-i}\boldsymbol{x}_i)(\boldsymbol{x}_i^\mathsf{T}\tilde{\boldsymbol{Q}}\boldsymbol{X}_{-i}\boldsymbol{v})\left(\delta - \frac{1}{p}\boldsymbol{x}_i^\mathsf{T}\boldsymbol{Q}_{-i}\boldsymbol{x}_i\right)\right]\right|$$

$$\leq \left|\frac{1}{p}\sum_{i=1}^n \left(\mathbb{E}\left[\left(\frac{1}{\sqrt{p}}\boldsymbol{u}^\mathsf{T}\boldsymbol{X}_{-i}^\mathsf{T}\boldsymbol{Q}_{-i}\boldsymbol{x}_i\right)^3\right]\mathbb{E}\left[\left(\frac{1}{\sqrt{p}}\boldsymbol{x}_i^\mathsf{T}\tilde{\boldsymbol{Q}}\boldsymbol{X}_{-i}\boldsymbol{v}\right)^3\right]\mathbb{E}\left[\left(\delta - \frac{1}{p}\boldsymbol{x}_i^\mathsf{T}\boldsymbol{Q}_{-i}\boldsymbol{x}_i\right)^3\right]\right)^{\frac{1}{3}}\right|$$

$$= \mathcal{O}\left(\frac{1}{\sqrt{p}}\right)$$

Similarly, we can show that:

$$\left|\frac{1}{p^2}\sum_{i=1}^n \mathbb{E}\left[\frac{u_i\boldsymbol{x}_i^\mathsf{T}\boldsymbol{Q}_{-i}\boldsymbol{C}\tilde{\boldsymbol{Q}}\boldsymbol{X}_{-i}\boldsymbol{v}}{1+\delta} + \frac{v_i\boldsymbol{u}^\mathsf{T}\boldsymbol{X}_{-i}^\mathsf{T}\boldsymbol{Q}_{-i}\boldsymbol{C}\tilde{\boldsymbol{Q}}\boldsymbol{x}_i}{1+\delta}\right.\right.$$

$$\left.\left. + u_iv_i\frac{\boldsymbol{x}_i^\mathsf{T}\boldsymbol{Q}_{-i}\boldsymbol{C}\tilde{\boldsymbol{Q}}\boldsymbol{x}_i}{1+\delta} - \frac{1}{p}\frac{\boldsymbol{u}^\mathsf{T}\boldsymbol{X}^\mathsf{T}\boldsymbol{Q}_{-i}\boldsymbol{x}_i\boldsymbol{x}_i^\mathsf{T}\boldsymbol{Q}\boldsymbol{C}\tilde{\boldsymbol{Q}}\boldsymbol{X}\boldsymbol{v}}{1+\delta}\right]\right| = \mathcal{O}\left(\frac{1}{\sqrt{p}}\right)$$

Finally, the remaining terms in $\Delta$ can be estimated as follows:

$$\Delta = \frac{1}{p^2} \sum_{i=1}^n \mathbb{E}\left[ -\frac{\boldsymbol{u}_i \boldsymbol{x}_i^\mathsf{T} \boldsymbol{Q}_{-i} \boldsymbol{x}_i \boldsymbol{x}_i^\mathsf{T} \tilde{\boldsymbol{Q}} \boldsymbol{X}_{-i} \boldsymbol{v}}{1 + \frac{1}{p} \boldsymbol{x}_i^\mathsf{T} \boldsymbol{Q}_{-i} \boldsymbol{x}_i} \right.$$

$$\left. - \frac{\boldsymbol{v}_i \boldsymbol{u}^\mathsf{T} \boldsymbol{X}_{-i}^\mathsf{T} \boldsymbol{Q}_{-i} \boldsymbol{x}_i \boldsymbol{x}_i^\mathsf{T} \tilde{\boldsymbol{Q}} \boldsymbol{x}_i}{1 + \frac{1}{p} \boldsymbol{x}_i^\mathsf{T} \boldsymbol{Q} \boldsymbol{x}_i} - \boldsymbol{u}_i \boldsymbol{v}_i \frac{\boldsymbol{x}_i^\mathsf{T} \boldsymbol{Q}_{-i} \boldsymbol{x}_i \boldsymbol{x}_i^\mathsf{T} \tilde{\boldsymbol{Q}} \boldsymbol{x}_i}{1 + \frac{1}{p} \boldsymbol{x}_i^\mathsf{T} \boldsymbol{Q}_{-i} \boldsymbol{x}_i} \right] + \mathcal{O}\left(\frac{1}{\sqrt{p}}\right)$$

$$= -\frac{2}{p} \frac{\delta \boldsymbol{u}^\mathsf{T} \mathbf{1} \bar{\boldsymbol{x}}^\mathsf{T} \tilde{\boldsymbol{Q}} \bar{\boldsymbol{x}} \mathbf{1}^\mathsf{T} \boldsymbol{v}}{1 + \delta} - \frac{\delta^2 \boldsymbol{u}^\mathsf{T} \boldsymbol{v}}{1 + \delta} + \mathcal{O}\left(\sqrt{\frac{\log p}{p}}\right)$$

Where the last equality is obtained through the following estimation:

$$\frac{1}{p^2} \sum_{i=1}^n \mathbb{E}\left[ \frac{\boldsymbol{v}_i \boldsymbol{u}^\mathsf{T} \boldsymbol{X}_{-i}^\mathsf{T} \boldsymbol{Q}_{-i} \boldsymbol{x}_i \boldsymbol{x}_i^\mathsf{T} \tilde{\boldsymbol{Q}} \boldsymbol{x}_i}{1 + \frac{1}{p} \boldsymbol{x}_i^\mathsf{T} \boldsymbol{Q}_{-i} \boldsymbol{x}_i} \right] = \frac{1}{p} \sum_{i=1}^n \mathbb{E}\left[ \frac{\boldsymbol{v}_i \boldsymbol{u}^\mathsf{T} \boldsymbol{X}_{-i}^\mathsf{T} \boldsymbol{Q}_{-i} \boldsymbol{x}_i \left(\frac{1}{p} \boldsymbol{x}_i^\mathsf{T} \tilde{\boldsymbol{Q}} \boldsymbol{x}_i (1 + \delta) - \delta \left(1 + \frac{1}{p} \boldsymbol{x}_i^\mathsf{T} \tilde{\boldsymbol{Q}} \boldsymbol{x}_i\right)\right)}{\left(1 + \frac{1}{p} \boldsymbol{x}_i^\mathsf{T} \boldsymbol{Q}_{-i} \boldsymbol{x}_i\right)(1 + \delta)} \right]$$

$$+ \frac{1}{p} \sum_{i=1}^n \frac{\boldsymbol{v}_i \delta \mathbb{E}[\boldsymbol{u}^\mathsf{T} \boldsymbol{X}_{-i}^\mathsf{T} \boldsymbol{Q}_{-i} \boldsymbol{x}_i]}{(1 + \delta)}$$

With the following bound:

$$\left| \frac{1}{p} \boldsymbol{x}_i^\mathsf{T} \tilde{\boldsymbol{Q}} \boldsymbol{x}_i (1 + \delta) - \delta \left(1 + \frac{1}{p} \boldsymbol{x}_i^\mathsf{T} \tilde{\boldsymbol{Q}} \boldsymbol{x}_i\right) \right|$$

$$= \left| \frac{1}{p} \boldsymbol{x}_i^\mathsf{T} \tilde{\boldsymbol{Q}} \boldsymbol{x}_i (1 + \delta) - \delta(1 + \delta) + \delta(1 + \delta) - \delta \left(1 + \frac{1}{p} \boldsymbol{x}_i^\mathsf{T} \tilde{\boldsymbol{Q}} \boldsymbol{x}_i\right) \right|$$

$$\leq \left| \frac{1}{p} \boldsymbol{x}_i^\mathsf{T} \tilde{\boldsymbol{Q}} \boldsymbol{x}_i - \delta \right| (1 + 2\delta),$$

we have again with Holder's inequality and Proposition A.4:

$$\frac{1}{p^2} \sum_{i=1}^n \mathbb{E}\left[ \frac{\boldsymbol{v}_i \boldsymbol{u}^\mathsf{T} \boldsymbol{X}_{-i}^\mathsf{T} \boldsymbol{Q}_{-i} \boldsymbol{x}_i \boldsymbol{x}_i^\mathsf{T} \tilde{\boldsymbol{Q}} \boldsymbol{x}_i}{1 + \frac{1}{p} \boldsymbol{x}_i^\mathsf{T} \boldsymbol{Q} \boldsymbol{x}_i} \right] = \frac{1}{p} \sum_{i=1}^n \frac{\boldsymbol{v}_i \delta \boldsymbol{u}^\mathsf{T} \mathbf{1} \bar{\boldsymbol{x}}^\mathsf{T} \tilde{\boldsymbol{Q}} \bar{\boldsymbol{x}}}{1 + \delta} + \mathcal{O}\left(\sqrt{\frac{\log p}{p}}\right)$$

Now that we estimated $\Delta$, it remains to estimate $\mathbb{E}[\frac{1}{p} \boldsymbol{X}^\mathsf{T} \tilde{\boldsymbol{Q}} \boldsymbol{X}]$. Indeed, given two unit norm vectors $u, v \in \mathbb{R}^n$ we have:

$$\mathbb{E}\left[ \frac{1}{p} \boldsymbol{u}^\mathsf{T} \boldsymbol{X}^\mathsf{T} \tilde{\boldsymbol{Q}} \boldsymbol{X} \boldsymbol{v} \right] = \frac{1}{p} \sum_{i,j=1}^n \boldsymbol{u}_i \boldsymbol{v}_j \mathbb{E}[\boldsymbol{x}_i^\mathsf{T} \tilde{\boldsymbol{Q}} \boldsymbol{x}_j] = \frac{1}{p} \sum_{i=1}^n \sum_{\substack{j=1 \\ j \neq i}}^n \boldsymbol{u}_i \boldsymbol{v}_j \bar{\boldsymbol{x}}^\mathsf{T} \tilde{\boldsymbol{Q}} \bar{\boldsymbol{x}} + \sum_{i=1}^n \boldsymbol{u}_i \boldsymbol{v}_i \delta$$

$$= \frac{1}{p} \bar{\boldsymbol{x}}^\mathsf{T} \tilde{\boldsymbol{Q}} \bar{\boldsymbol{x}} \boldsymbol{u}^\mathsf{T} \mathbf{1} \mathbf{1}^\mathsf{T} \boldsymbol{v} + (\delta - \frac{1}{p} \bar{\boldsymbol{x}}^\mathsf{T} \tilde{\boldsymbol{Q}} \bar{\boldsymbol{x}}) \boldsymbol{u}^\mathsf{T} \boldsymbol{v} = \frac{1}{p} \bar{\boldsymbol{x}}^\mathsf{T} \tilde{\boldsymbol{Q}} \bar{\boldsymbol{x}} \boldsymbol{u}^\mathsf{T} \boldsymbol{M}_1 \boldsymbol{v}^\mathsf{T} + \delta \boldsymbol{u}^\mathsf{T} \boldsymbol{v} + \mathcal{O}\left(\frac{1}{p}\right)$$

since we have $\bar{\boldsymbol{x}}^\mathsf{T} \tilde{\boldsymbol{Q}} \bar{\boldsymbol{x}} = \mathcal{O}(1)$ by Lemma A.2; we introduced the matrix $\boldsymbol{M}_1 = \mathbf{1}\mathbf{1}^\mathsf{T}$. Therefore we have the following estimation:

$$\frac{1}{p} \mathbb{E}\left[\boldsymbol{X}^\mathsf{T} \boldsymbol{Q} \boldsymbol{X}\right] = \frac{\delta}{1 + \delta} \boldsymbol{I}_n + \frac{1}{p}\left(\frac{1 - \delta}{1 + \delta}\right) \bar{\boldsymbol{x}}^\mathsf{T} \tilde{\boldsymbol{Q}} \bar{\boldsymbol{x}} \boldsymbol{M}_1 + \mathcal{O}_{\|\cdot\|}\left(\sqrt{\frac{\log p}{p}}\right)$$

where $\boldsymbol{A} = \boldsymbol{B} + \mathcal{O}_{\|\cdot\|}(\alpha(p))$ means that $\|\boldsymbol{A} - \boldsymbol{B}\| = \mathcal{O}(\alpha(p))$. Finally, since $\boldsymbol{R}$ concentrates around its mean, we can then conclude:

$$\boldsymbol{R} = \frac{1}{z}\left(\boldsymbol{I}_n - \frac{1}{p} \boldsymbol{X}^\mathsf{T} \boldsymbol{Q} \boldsymbol{X}\right) = \frac{1}{z} \frac{1}{1 + \delta} \boldsymbol{I}_n + \frac{\delta - 1}{pz(\delta + 1)} \bar{\boldsymbol{x}}^\mathsf{T} \tilde{\boldsymbol{Q}} \bar{\boldsymbol{x}} \boldsymbol{M}_1 + \mathcal{O}_{\|\cdot\|}\left(\sqrt{\frac{\log p}{p}}\right).$$

$\square$

# B  PROOF OF PROPOSITION 3.1

*Proof.* Since the Lipschitz constant of a composition of Lipschitz functions is bounded by the product of their Lipschitz constants, we consider the case $N = 1$ and a linear activation function. In this case, the Lipschitz constant corresponds to the largest singular value of the weight matrix. We consider the following notations for the proof

$$\bar{\boldsymbol{W}}_t = \boldsymbol{W}_t - \eta \boldsymbol{E}_t \text{ with } [\boldsymbol{E}_t]_{i,j} \sim \mathcal{N}(0,1)$$
$$\boldsymbol{W}_{t+1} = \bar{\boldsymbol{W}}_t - \max(0, \bar{\sigma}_{1,t} - \sigma_*) \, \bar{\boldsymbol{u}}_{1,t} \bar{\boldsymbol{v}}_{1,t}^\mathsf{T}$$

where $\bar{\sigma}_{1,t} = \sigma_1(\bar{\boldsymbol{W}}_t)$, $\bar{\boldsymbol{u}}_{1,t} = \boldsymbol{u}_1(\bar{\boldsymbol{W}}_t)$ and $\bar{\boldsymbol{v}}_{1,t} = \boldsymbol{v}_1(\bar{\boldsymbol{W}}_t)$. The effect of spectral normalization is observed in the case where $\sigma_* > \bar{\sigma}_{1,t}$, otherwise the Lipschitz constant is bounded by $\sigma_*$. We therefore have

$$\|\bar{\boldsymbol{W}}_t\|_F^2 \leq \|\boldsymbol{W}_t\|_F^2 + \eta^2 d_1 d_0 \tag{16}$$
$$\|\boldsymbol{W}_{t+1}\|_F^2 = \|\bar{\boldsymbol{W}}_t\|_F^2 + \sigma_*^2 - \bar{\sigma}_{1,t}^2 \tag{17}$$

- If $\|\boldsymbol{W}_{t+1}\|_F \geq \|\boldsymbol{W}_t\|_F$, we have by equation 16 and equation 17

  $$\|\bar{\boldsymbol{W}}_t\|_F^2 \leq \|\bar{\boldsymbol{W}}_t\|_F^2 + \sigma_*^2 - \bar{\sigma}_{1,t}^2 + \eta^2 d_1 d_0 \quad \Rightarrow \quad \|\bar{\boldsymbol{W}}_t\| = \bar{\sigma}_{1,t} \leq \sqrt{\sigma_*^2 + \eta^2 d_1 d_0} = \delta$$

  And since $\|\boldsymbol{W}_{t+1}\| \leq \|\bar{\boldsymbol{W}}_t\|$, we have $\|\boldsymbol{W}_{t+1}\| \leq \delta$.

- Otherwise, if there exits $\tau$ such that $\|\boldsymbol{W}_{\tau+1}\|_F < \|\boldsymbol{W}_\tau\|_F$, then for all $\varepsilon > 0$ there exists an iteration $\tau' \geq \tau$ such that $\|\boldsymbol{W}_{\tau'}\| \leq \delta + \varepsilon$. Indeed, otherwise we denote $\varepsilon_t = \|\boldsymbol{W}_t\|^2 - \delta^2$ and $\varepsilon_t > 0$ for all $t \geq \tau$. And if for all $t \geq \tau$, $\|\boldsymbol{W}_{t+1}\|_F \leq \|\boldsymbol{W}_t\|_F$, we have by equation 16 and equation 17

  $$\|\boldsymbol{W}_t\|_F^2 - \|\boldsymbol{W}_{t+1}\|_F^2 \geq \|\bar{\boldsymbol{W}}_t\|^2 - \delta^2 \geq \|\boldsymbol{W}_{t+1}\|^2 - \delta^2 = \varepsilon_{t+1}$$

  Integrating the above expression from $\tau$ to $T - 1 \geq \tau$, we end up with

  $$\|\boldsymbol{W}_\tau\|_F^2 - \|\boldsymbol{W}_T\|_F^2 \geq \sum_{t=\tau}^{T-1} \varepsilon_t \quad \Rightarrow \quad 0 \leq \|\boldsymbol{W}_T\|_F^2 \leq \|\boldsymbol{W}_\tau\|_F^2 - \sum_{t=\tau}^{T-1} \varepsilon_t,$$

  therefore, when $T \to \infty$, $\varepsilon_t$ has to tend to 0 otherwise the right hand-side of the last inequality will tend to $-\infty$ which is absurd.

  □

