# OpenReview forum: "Random Matrix Theory Proves that Deep Learning Representations of GAN-data Behave as Gaussian Mixtures"
_ICLR.cc/2020/Conference — Reject_

### Official Review · AnonReviewer2 · 2019-10-20
**Official Blind Review #2**

**Rating:** 6

**Review:**

The authors generalize Gaussian random vectors to a broader class of "concentrated" vectors which they use as their primary tool for analysis of the latent representations learned by GANs. They show that the spectral behavior (i.e. spectra and leading eigenspaces) of the Gram matrix computed over GAN representations is the same as those produced by a high dimensional Gaussian Mixture Models (GMMs). Furthermore, they show that the "sufficient statistics" (i.e. measures of information encoded in the latent representations) for GANs depend only on their first and second moments. Thus, for data that follows Gaussian mixture patterns, GANs and GMMs behave identically. The authors also show that common neural network operations (linear/convolutional layers, pooling, batch normalization, ReLU activations) are Lipschitz transformations, where the Lipschitz constants can be bounded using spectral normalization (this is borrowed from previous work). They also provide some empirical analysis to verify their theoretical findings.

Overall, the paper is well organized and the theoretical results are both compelling and thorough. The experimental results also follow nicely from the theory. Admittedly, this reviewer is not well versed enough in this area of mathematics to provide thorough critical insight about the derivations and proofs. However, the notation is clear and the general arguments appear to be sound. The authors' theoretical results are significant, and provide a much needed step forward in formalizing and understanding deep generative models.

**Experience Assessment:**

I do not know much about this area.

**Review Assessment: Checking Correctness Of Derivations And Theory:**

I assessed the sensibility of the derivations and theory.

**Review Assessment: Checking Correctness Of Experiments:**

I assessed the sensibility of the experiments.

**Review Assessment: Thoroughness In Paper Reading:**

I made a quick assessment of this paper.

---

> ### Author Response · Authors · 2019-11-13
> **Response to Reviewer 2**
>
> We thank the reviewer for his time reviewing our work and for his comments on our paper.

---

### Official Review · AnonReviewer1 · 2019-10-22
**Official Blind Review #1**

**Rating:** 3

**Review:**

This paper provides a formal proof that the data produced by a GAN are concentrated vectors. The proof is rather intuitive: a GAN essentially consists of a Lipschitz function that takes as input a latent vector drawn from a Gaussian distribution and therefore one can use Lipschitz concentration bounds to prove that the transformed output satisfies a concentration property. Overall, I think this is a very intuitive idea and I fail to see the added value of such an observation. I provide detailed feedback below, I would really like to get a sensible answer regarding the novelty aspect.

Gaussian assumption
It is of course very common to sample the latent vectors from a Gaussian distribution, although one could potentially use a different distribution (uniform, gamma, …). To what extent could these results extend to other distributions?

Prior work
The paper does not appropriately discuss prior work that impose a Lipschitz constraint while training, see e.g.:
Gulrajani, Ishaan, et al. "Improved training of wasserstein gans." Advances in neural information processing systems. 2017.
Roth, Kevin, et al. "Stabilizing training of generative adversarial networks through regularization." Advances in neural information processing systems. 2017.

Theorem 3.3
I would like further clarifications regarding this theorem. This theorem is essentially a concentration bound for the resolvent of the gram matrix G around its mean. The expectation is already computed in (Louart & Couillet, 2019) so the contribution in this theorem seems to be in showing that the result only depends on the first and second order statistics. You claim this is a surprising result, although it does not seem so surprising to me given that this is an asymptotic result. Could you comment on deriving a non-asymptotic result instead?

Added value
As mentioned earlier, I do not see what particular insight is this paper bringing. There are some obvious connections that one could make, e.g. implications such as robustness to adversarial samples depending on the Lipschitz constant, or perhaps improve generalization property, but some of these connections are already made in prior work (see comment above). What particular insight do we get from your analysis?

Experiments
I think a more detailed study regarding the effect of various regularization schemes would be valuable. One could for instance compare various networks with/without batchnorm, resnet connections, ...


**Experience Assessment:**

I have published in this field for several years.

**Review Assessment: Checking Correctness Of Derivations And Theory:**

I carefully checked the derivations and theory.

**Review Assessment: Checking Correctness Of Experiments:**

I assessed the sensibility of the experiments.

**Review Assessment: Thoroughness In Paper Reading:**

I read the paper thoroughly.

---

> ### Author Response · Authors · 2019-11-13
> **Response to Reviewer 1**
>
> We first thank the reviewer for the time taken to review our paper and for his pertinent and constructive comments.
>
> As hinted by the reviewer, the aim of our article is to show that data produced by GANs are concentrated vectors by design and further by analyzing the spectral behavior of the Gram matrix for a mixture of concentrated vectors, we show that only first and second order class-wise statistics of these vectors matter to describe the spectral behavior of the Gram matrix.
>
> Fundamentally, this result suggests that linear classifiers (which rely on the Gram matrix) retrieve the information from GAN data (their DL representations or GAN data themselves) only through their class-wise means and covariances. From a practical standpoint, if we can assimilate real data (e.g. images) to data produced by GANs, our result suggests that DL representations of real data can be described by simple GMMs for linear models, one can thus exploit this result to choose the “best” representation network among a set of representation networks given a certain learning task (in a transfer learning scheme for instance), only by exploiting consistent functionals of the first and second-order statistics of the representations. These questions are currently under investigation and shall be provided in a future study given our actual findings. We further answer point by point to the reviewer's questions below.
>
> Gaussian assumption:
> The reviewer asked about the extension of our results to other choices for the prior instead of the Gaussian distribution. Note that our results hold for any choice of the prior for which the sampled latent vectors are concentrated vectors, this is the case for random Gaussian vectors as we have shown in Proposition 2.2, it holds also for random vectors with uniformly distributed entries thanks to Proposition 2.8 of (Ledoux, 2005) and one can show the concentration of random vectors sampled from the Gamma distribution thanks to Theorem 2.7 of (Ledoux, 2005) and more distributions (see (Ledoux, 2005) and the references therein). We have chosen to work under the Gaussian distribution as a prior in our paper since it is the most commonly used to sample the latent vectors.
>
> Prior work:
> Although this is not the main result of our paper, the reviewer pointed out other prior work concerning Lipschitz constraints during neural networks training. We have discussed these works as suggested by the reviewer and we thank him for the provided references.
>
> Theorem 3.3:
> This theorem constitutes our main theoretical result and provides a concentration bound for the resolvent of the Gram matrix G around its mean and a deterministic estimator of its mean (called deterministic equivalent in the Random Matrix Theory literature). Louart & Couillet, 2019 have studied the resolvent of the sample covariance matrix, whereas our paper studies the resolvent of the Gram matrix, it should be noted that the result for the Gram matrix is not straightforward given the concentration of the sample covariance matrix, and technically speaking one needs to handle the statistical dependencies between the data matrix X and the resolvent Q of the sample covariance matrix in the expression X’QX (see proof of Theorem 3.3). We have notably handled these dependencies tanks to the Propositions A.3 and A.4 of appendix A.
>
> On the other side, concerning the reviewer’s comment about deriving a non-asymptotic result, it should be noted that our result is usually called a "quasi-asymptotic result" since it takes into account the dependence between p and n (n = O(p)) through the bound O(√log(p)/p), usually the rate of convergence is not given and it is just set that lim ∥∥ER(z)− ̃R(z)∥ = 0 (this is then an "asymptotic result"). Depending on the level of approximation one is looking for, one can derive any non-asymptotic result. The constant in O(√log(p)/p) is not given because it is ugly and not optimized, in particular, it depends on the constant of concentration σ given in Definition 2.1. The difficulty holds in the fact that n AND p are jointly going to infinity which makes the result not so easy to show under our assumptions. Moreover, since we derived our result on the assumption that both p and n are large, our findings describe the spectral behavior of the Gram matrix for p and n large enough and of the same order as depicted in Figure 4 (where n=1500 and p \in {2048, 4096, 1920}).

---

> ### Author Response · Authors · 2019-11-13
> **Additional response to Reviewer 1**
>
> Added value:
> The reviewer questioned about our main findings. As we have discussed on top of this comment, our analysis is based on the observation that DL representations of data generated by GANs are concentrated vectors by design, and further analyzing the spectral properties of the Gram matrix for concentrated vectors, we have shown that only first and second class-wise moments of concentrated vectors describe the behavior of the Gram matrix computed on these vectors. This result brings the insight that, for GAN data, linear models (which rely on the Gram matrix) retrieve the information from data only through the class-wise means and covariances (this is, in particular, the case for the last layer of a neural network). Furthermore, if we can assimilate real images to images produced by GANs, therefore their DL representations will behave as a GMM for linear models. This result suggests that one could select the most relevant DL representation (among a set of given DL representations) for a given learning task based only on simple functionals of the class-wise means and covariances. This study will be provided in a future work given our actual findings, in particular, through the establishment of the above-mentioned functionals and their consistent estimation with a few amounts of data.
>
> Experiments:
> The main purpose of our experiments is to show that the spectral behavior of the Gram matrix is the same for DL representations of GAN data as for a GMM model with the same first and second moments computed on the representations, as proved in Theorem 3.3 since they are concentrated vectors by design. We have shown that this result extends to real images as they can be assimilated to GAN-generated ones. We have particularly considered pre-trained representation networks (Resnet, VGG and Densenet), and thus considering different regularization schemes will affect the quality of the considered representations and this will be considered in a future study since it not the main scope of the actual paper.

---

### Official Review · AnonReviewer3 · 2019-10-23
**Official Blind Review #3**

**Rating:** 1

**Review:**

In this paper, the authors claim to establish a Lipschitz bound on neural networks, initialized randomly, and trained until convergence.  They also claim to establish probabilistic concentration of the resolvent of the Gram matrix from a mixture of k distributions with varying means and covariances.  The authors also present the spectrum and leading eigenspace of the Gram matrix for representations by CNNs of images generated by a GAN.

It is not clear to this reviewer what exactly the authors have proved and what significance it has.  For example, in Proposition 3.1, the authors talk about dynamics with a learning rate, where W <- W - \eta E where E has standard normal entries and \eta is a learning rate.  It is unclear what the learning problem is, or why the learning rate is modifying a random modification.  I understand the general idea that Lipschitz functions of random-quantities-that-enjoy-concentration-estimates also enjoy concentration estimates.  This is well known from the random matrix theory literature.  The authors need to make a claim about how this informs either the development or understanding of deep learning technologies.  The authors should consider what the implications of their results could be, and then do the research to establish that those implications hold up.

The authors claim that their results constitute "a first step towards the theoretical understanding of complex objects such as DL representations."  This claim is false.  This very conference is on Learning Representations, and presumably at least one paper in the past 7 years makes progress towards the theoretical understanding of Deep Learning representations.

Because of the overly bold claims with insufficient clarity and justification, I recommend that this paper be rejected.

**Experience Assessment:**

I have published one or two papers in this area.

**Review Assessment: Checking Correctness Of Derivations And Theory:**

I assessed the sensibility of the derivations and theory.

**Review Assessment: Checking Correctness Of Experiments:**

I assessed the sensibility of the experiments.

**Review Assessment: Thoroughness In Paper Reading:**

I read the paper at least twice and used my best judgement in assessing the paper.

---

> ### Author Response · Authors · 2019-11-13
> **Response to Reviewer 3**
>
> We thank the reviewer for the time spent reviewing our work. We believe that there might be a slight misunderstanding of the main consequences of our findings. Indeed, our work does not establish a Lipschitz bound on trained neural networks as understood by the reviewer.
>
> In contrast, our paper shows that DL representations of data generated by GANs are random vectors that fall within the class of concentrated vectors (see Definition 2.1). Relying on this observation, we have studied the spectral behavior of the Gram matrix for concentrated vectors, and we have shown that it has the same behavior as if data are sampled from a GMM in the high-dimensional setting (when both the number of samples and their dimension are of the same order). This means that linear models (which rely on the Gram matrix) retrieve the information from data based only on their class-wise means and covariances.
>
> As a result, If we can assimilate real images to GAN-generated ones, we know from our analysis that only first and second-order moments of their representations describe the classification accuracy of linear models, in that sense, one can choose the best representation network given a certain learning task based only on some functionals of the first and second moment statistics estimated on a small amount of data. We will provide this analysis in a future study based on our actual findings (see our comment to Reviewer 1 for more discussions).
>
> To summarize and clarify our findings to the reviewer, our paper shows that large Gram matrices of large concentrated vectors have the same spectral behavior as if the Gram matrices were computed on random Gaussian vectors with the same class-wise means and covariances. We have notably identified that DL representations of GAN-data fall within the class of concentrated vectors by construction, and we have validated our theoretical results by considering CNN representations (e.g., Resnet, VGG, Densenet) of GAN-generated images as shown in Figure 4, we further observed that our results extend to real images.
>
> Since we have characterized the behavior of the Gram matrix (i.e., linear models) for DL representations of GAN-data, and if we can assimilate real data to GAN generated ones, we understand in that sense that DL representations can be fully described by their first and second class-wise moments (sufficient statistics) for linear models, which describes the “quality” of the considered DL representation through the aforementioned sufficient statistics. In that sense, we believe that our results constitute a different approach towards the statistical understanding and modeling of DL representations, relying on random matrix theory and large dimensional statistics. We apologize for our overclaimed statement and we thank the Reviewer for pointing it out, we have corrected this statement in the revised version.

---

### Author Response · Authors · 2019-11-15
**Thanks to reviewers and ACs**

We gratefully thank all reviewers for their feedback and constructive comments, and also for the time taken to review our paper. We also thank the ACs for their efforts. We have tried to respond to all the reviewer's comments and have updated our paper by addressing them. We hope this will encourage reviewers 1 and 3 to reconsider their scores.

---

### Decision · Program_Chairs · 2019-12-19

**Decision:**

Reject

**Comment:**

The paper theoretically shows that the data (embedded by representations learned by GANs) are essentially the same as a high dimensional Gaussian mixture. The result is based on a recent result from random matrix theory on the covariance matrix of data, which the authors extend to a theorem on the Gram matrix of the data. The authors also provide a small experiment comparing the spectrum and principle 2D subspace of BigGAN and Gaussian mixtures, demonstrating that their theorem applies in practice.

Two of the reviews (with confident reviewers) were quite negative about the contributions of the paper, and the reviewers unfortunately did not participate in the discussion period.

Overall, the paper seems solid, but the reviews indicate that improvements are needed in the structure and presentation of the theoretical results. Given the large number of submissions at ICLR this year, the paper in its current form does not pass the quality threshold for acceptance.